# Astrocyte and L-lactate in the anterior cingulate cortex modulate schema memory and neuronal mitochondrial biogenesis

**Mastura Akter[1,2†], Mahadi Hasan[1,2], Aruna Surendran Ramkrishnan[1,2], Zafar Iqbal[1,2,3], Xianlin Zheng[1,2‡], Zhongqi Fu[1,3], Zhuogui Lei[1,2], Anwarul Karim[1§], Ying Li[1,2,3,4]\***

[1]Department of Neuroscience, City University of Hong Kong, Hong Kong SAR, China; [2]Department of Biomedical Sciences, City University of Hong Kong, Hong Kong SAR, China; [3]Centre for Regenerative Medicine and Health, Hong Kong Institute of Science & Innovation, Chinese Academy of Sciences, Hong Kong SAR, China; [4]Centre for Biosystems, Neuroscience, and Nanotechnology, City University of Hong Kong, Hong Kong SAR, China

**\*For correspondence:**
yingli@cityu.edu.hk

**Present address:** [†]Brown Foundation Institute of Molecular Medicine, University of Texas Health Science Center, Houston, Texas, United States; [‡]Krieger School of Arts & Sciences, Johns Hopkins University, Baltimore, Maryland, United States; [§]Department of Neurology, Baylor College of Medicine, Houston, Texas, United States

**Competing interest:** The authors declare that no competing interests exist.

**Abstract** Astrocyte-derived L-lactate was shown to confer beneficial effects on synaptic plasticity and cognitive functions. However, how astrocytic $G_i$ signaling in the anterior cingulate cortex (ACC) modulates L-lactate levels and schema memory is not clear. Here, using chemogenetic approach and well-established behavioral paradigm, we demonstrate that astrocytic $G_i$ pathway activation in the ACC causes significant impairments in flavor-place paired associates (PAs) learning, schema formation, and PA memory retrieval in rats. It also impairs new PA learning even if a prior associative schema exists. These impairments are mediated by decreased L-lactate in the ACC due to astrocytic $G_i$ activation. Concurrent exogenous L-lactate administration bilaterally into the ACC rescues these impairments. Furthermore, we show that the impaired schema memory formation is associated with a decreased neuronal mitochondrial biogenesis caused by decreased L-lactate level in the ACC upon astrocytic $G_i$ activation. Our study also reveals that L-lactate-mediated mitochondrial biogenesis is dependent on monocarboxylate transporter 2 (MCT2) and NMDA receptor activity – discovering a previously unrecognized signaling role of L-lactate. These findings expand our understanding of the role of astrocytes and L-lactate in the brain functions.

## Editor's evaluation

This is an important study that investigates the role of astrocytic Gi signaling in the anterior cingulate cortex in the modulation of extracellular L-lactate level and consequently impairment in flavor-place paired associates learning. The evidence supporting the authors' main conclusions is convincing. Additionally, the study provides compelling evidence to suggest the molecular mechanism by which astrocyte-produced L-lactate may influence mitochondrial biogenesis in neurons, thereby affecting schema memory. This study expands our understanding of how disruptions in astrocytic functions can impair cognitive processing and, therefore, could have clinical relevance.

## Introduction

Astrocyte, the predominant type of glia in the brain, is involved in complex brain functions including learning, memory, and synaptic plasticity (*Santello et al., 2019*; *Doron et al., 2022*). They can modulate neuronal activity by releasing and regulating different neuroactive molecules (*Ota et al., 2013*; *Doron et al., 2022*). Astrocytes express numerous transporters and receptors including G protein-coupled receptors (GPCRs) to modulate their own as well as neuronal activity. Designer receptors exclusively activated by designer drugs (DREADDs) are genetically modified GPCRs which allow researchers to control cellular activity via modulation of GPCR signaling with the application of selective ligands (*Urban and Roth, 2015*; *Yu et al., 2020*). These chemogenetic tools have been used to modulate the function of neurons and astrocytes in different brain regions (*Armbruster et al., 2007*; *Alexander et al., 2009*; *Zhu et al., 2014*; *Koike et al., 2016*; *Adamsky et al., 2018*; *Jones et al., 2018*; *Durkee et al., 2019*; *Pati et al., 2019*; *Kol et al., 2020*; *Oguchi et al., 2021*; *Lei et al., 2022*; *Liu et al., 2022*). Using chemogenetic approach, astrocytic $G_i$ pathway activation in the hippocampus has recently been shown to modulate cognitive functions (*Jones et al., 2018*; *Kol et al., 2020*; *Liu et al., 2022*), although the mechanism is still not fully understood. Moreover, how astrocytic $G_i$ signaling in the anterior cingulate cortex (ACC) affects cognitive functions – particularly schema memory – is yet unknown.

L-lactate is a metabolic end product of glycolysis and works as an energy substrate for various tissues, including the brain. According to the astrocyte-neuron L-lactate shuttle hypothesis (*Pellerin and Magistretti, 1994*; *Magistretti and Allaman, 2018*), L-lactate is produced by astrocytes through glycogenolysis and glycolysis and then transported into the neurons through monocarboxylate transporter 2 (MCT2) to fuel the high metabolic demand in neurons to maintain various physiological activities including neural plasticity and memory formation (*Magistretti and Allaman, 2018*). However, a recent study has argued that the energy demand during neuronal activation is fueled by glucose rather than astrocyte-derived L-lactate (*Díaz-García et al., 2017*). Nevertheless, multiple studies clearly demonstrated that L-lactate confers beneficial effect in learning and memory (*Newman et al., 2011*; *Suzuki et al., 2011*; *Wang et al., 2017*; *Harris et al., 2019*; *Netzahualcoyotzi and Pellerin, 2020*; *Vezzoli et al., 2020*; *Iqbal et al., 2022*). Administration of L-lactate into the hippocampus enhanced memory in rats whereas inhibition of astrocytic glycogenolysis or inhibition of astrocytic or neuronal MCTs in the hippocampus impaired memory formation (*Newman et al., 2011*; *Suzuki et al., 2011*; *Netzahualcoyotzi and Pellerin, 2020*). However, how L-lactate in the ACC affects schema memory is yet unknown.

Using a behavioral paradigm, Tse et al. showed that learning of multiple flavor-place paired associates (PAs) leads to the development of cortical associative schema in rats that allows rapid assimilation of new PAs (NPAs) into the existing schema (*Tse et al., 2007*; *Tse et al., 2011*). Previously, our team showed that bilateral infusion of lidocaine (a neuronal blocker) into the hippocampus or the ACC prevents PA learning, schema formation, and memory retrieval (*Hasan et al., 2019*). The study also showed increased oligodendrogenesis and adaptive myelination in the ACC of rats after repeated PA training. Furthermore, it demonstrated that myelination in the ACC is necessary for PA learning and memory retrieval, suggesting an important role of oligodendrocytes in schema memory. However, the role of astrocytes in the ACC in PA learning, schema formation, and memory retrieval is still unknown. Using hM4Di (a $G_i$-coupled GPCR) DREADD, here we show that astrocytic $G_i$ pathway activation in the ACC causes significant impairments in PA learning, schema formation, and memory retrieval in rats. We also show that the impairments are mediated by a decrease in L-lactate level in the ACC upon astrocytic $G_i$ activation. Concurrent exogenous L-lactate administration into the ACC rescues these impairments. Furthermore, we discover that astrocytic $G_i$ activation diminishes neuronal mitochondrial biogenesis, which could be rescued by exogenous L-lactate, and that L-lactate-induced neuronal mitochondrial biogenesis requires MCT2 and NMDAR activity – revealing a previously unrecognized L-lactate signaling mechanism in controlling neuronal mitochondrial biogenesis.

## Results

### Expression of hM4Di in the ACC astrocytes

The ACC of both sides were injected with adeno-associated virus serotype 8 (AAV8) vector encoding mCherry-tagged hM4Di under the control of glial fibrillary acidic protein (GFAP) promoter to drive

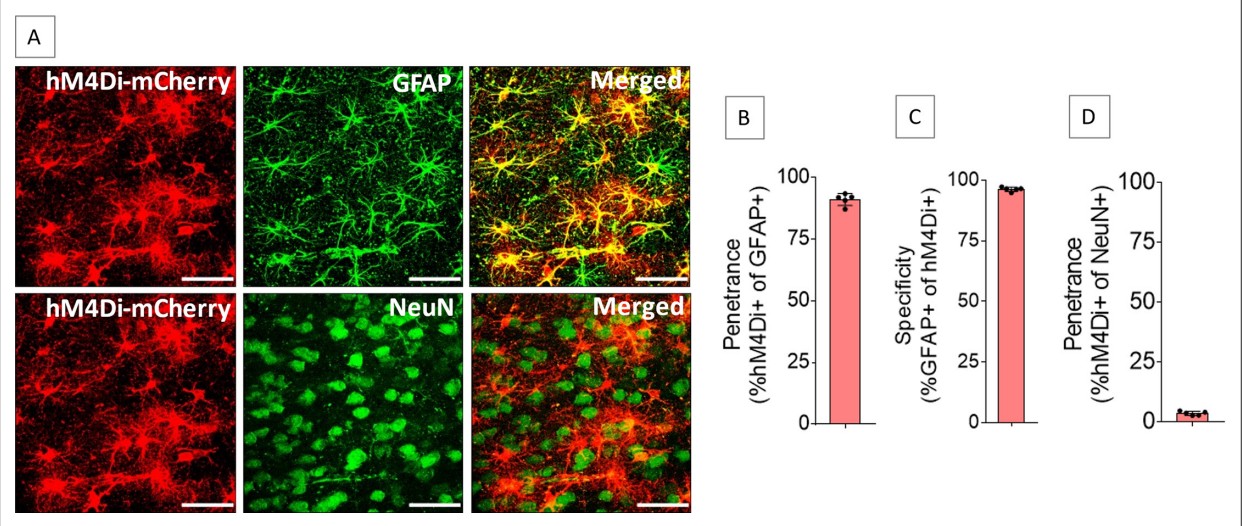

**Figure 1.** Expression of hM4Di in anterior cingulate cortex (ACC) astrocytes. Injection of AAV8-GFAP-hM4Di-mCherry into the ACC resulted in expression of hM4Di (**A**) in 91.1 ± 2.4% of GFAP-positive cells (**B**) with 96.3 ± 0.9% specificity (**C**), whereas 3.6 ± 0.8% of NeuN-positive cells expressed hM4Di (**D**). n=5 rats (3 sections/rat). Scale bars: 50 µm.

The online version of this article includes the following source data and figure supplement(s) for figure 1:

**Source data 1.** Zip file containing data for *Figure 1B–D* in GraphPad Prism file format.

**Figure supplement 1.** Microinjection and cannulation in the anterior cingulate cortex (ACC).

hM4Di expression in the ACC astrocytes (AAV8-GFAP-hM4Di-mCherry). The hM4Di is a modified human muscarinic receptor M4 that has been engineered to be insensitive to the endogenous ligand acetylcholine but can be activated by its selective ligand clozapine-*N*-oxide (CNO) (*Armbruster et al., 2007*). Injection of AAV8-GFAP-hM4Di-mCherry into the ACC resulted in expression of hM4Di in the ACC astrocytes (*Figure 1A*) with high penetrance (91.1 ± 2.4%, *Figure 1B*) and specificity (96.3 ± 0.9%, *Figure 1C*). Penetrance in NeuN-positive cells was low (3.6 ± 0.8%, *Figure 1D*).

## G$_i$ pathway activation in the ACC astrocytes impairs PA learning

PA learning is hippocampus-dependent. Training rats with several PAs leads to schema formation, which is stored in the ACC, and the learned PAs gradually become hippocampus-independent (*Tse et al., 2007*; *Tse et al., 2011*; *Hasan et al., 2019*; *Liu et al., 2022*). Astrocytic G$_i$ pathway activation has been shown to modulate different cognitive functions (*Jones et al., 2018*; *Kol et al., 2020*; *Liu et al., 2022*). However, the effect of the ACC astrocytic G$_i$ activation on schema memory is yet unknown. To investigate this, bilateral injection of AAV8-GFAP-hM4Di-mCherry into the ACC of rats (n=15) was done to express hM4Di-mCherry in the astrocytes. This group of rats received intraperitoneal (IP) CNO (3 mg/kg body weight) 30 min before the start of each session and 30 min after the end of each session. Hereafter, this group will be referred to as the 'hM4Di-CNO group'. Another group of rats (n=8) was used as control which did not receive AAV8-GFAP-hM4Di-mCherry injection or CNO.

After habituation and pretraining, we trained both groups of rats with six PAs (sessions 1, 2, 4–8, 10–17) (*Figure 2A*). Control group showed a gradual increase in performance index (PI) throughout the PA sessions (*Figure 2A*). At S6, the PI was significantly increased above the chance level (62.5 ± 5.3%, one-sample t-test with hypothetical value of 50%, t=6.71, df = 7, p<0.001) and it remained above the chance level throughout the following PA sessions. At S17, the PI reached the maximum level (77.6 ± 4.6%). This result is consistent with previous reports (*Tse et al., 2007*; *Hasan et al., 2019*; *Liu et al., 2022*). However, the hM4Di-CNO group consistently had lower PI (*Figure 2A*) compared to the control group in all PA training sessions from S6 to S17 (statistical data is given in *Supplementary file 1*). At S6, the PI of this group was 49.1 ± 5.4% and at S17 it was 61.3 ± 5%. Consistently, the PI of this group was 50.2 ± 7.4% when two NPAs (NPAs 7 and 8) were introduced at S19, whereas it was significantly higher (69.5 ± 7.1%) in the control group (p<0.001, unpaired t-test, t ratio = 6.1, df = 21). These findings indicated that G$_i$ pathway activation in the ACC astrocytes during and immediately after

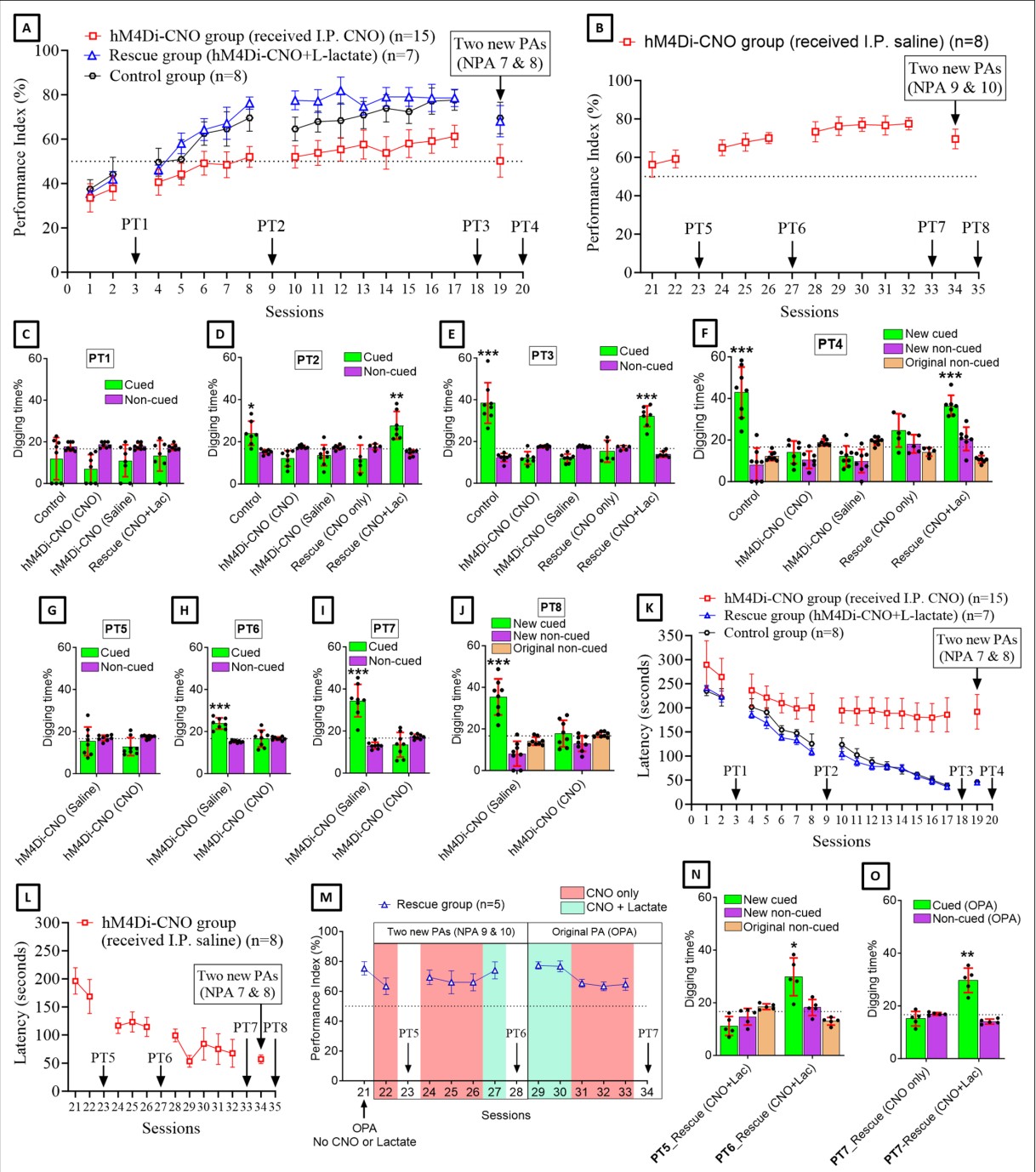

**Figure 2.** Astrocytic $G_i$ pathway activation in the anterior cingulate cortex (ACC) impairs paired associate (PA) learning, schema consolidation, memory retrieval, and assimilation of new PAs (NPAs) into existing schema whereas L-lactate can rescue these impairments. (**A**) Performance index (PI) (mean ± SD) during the acquisition of the six original PAs (OPAs) (S1–2, 4–8, 10–17) and NPAs (S19) of the control (n=8), hM4Di-CNO (n=15), and rescue (hM4Di-CNO+L-lactate) (n=7) groups. From S6 onward, hM4Di-CNO group consistently showed lower PI compared to control. However, concurrent L-lactate administration into the ACC (rescue group) can rescue this impairment. (**B**) PI (mean ± SD) of hM4Di-CNO group (n=8) from S21 onward showing gradual increase in PI when CNO was withdrawn. (**C, D, and E**) Non-rewarded PTs (PT1, PT2, and PT3 conducted on S3, S9, and S18, respectively) to test memory retrieval of OPAs for the control, hM4Di-CNO, and rescue groups. The percentage of digging time at the cued location relative to that at the non-cued locations are shown (mean ± SD). In both PT2 and PT3, the control group spent significantly more time digging the cued sand well above the chance level, indicating that the rats learned OPAs and could retrieve it. Contrasting to this, hM4Di-CNO group did not spend more time digging the cued sand well above the chance level irrespective of CNO administration before the PTs. The rescue group showed results similar to the hM4Di-CNO group if CNO is given without L-lactate. On the other hand, they showed results similar to the control group if L-lactate is concurrently given with

*Figure 2 continued on next page*

*Figure 2 continued*

CNO, indicating that this group learned OPAs and could retrieve it. *p<0.05, **p<0.01, ***p<0.001, one-sample t-test comparing the proportion of digging time at the cued sand well with the chance level of 16.67%. Non-rewarded PT4 (S20) which was conducted after replacing two OPAs with two NPAs (NPAs 7 and 8) in S19 for the control, hM4Di-CNO, and rescue groups. Results show that the control group spent significantly more time digging the new cued sand well above the chance level indicating that the rats learned the NPAs from S19 and could retrieve it in this PT. Contrasting to this, hM4Di-CNO group did not spend more time digging the new cued sand well above the chance level irrespective of CNO administration before the PT. The rescue group showed results similar to the hM4Di-CNO group if CNO is given without L-lactate. On the other hand, they showed results similar to the control group if L-lactate is concurrently given with CNO indicating that this group learned NPAs from S19 and could retrieve it. ***p<0.001, one-sample t-test comparing the proportion of digging time at the new cued sand well with the chance level of 16.67%. (**G, H, and I**) Non-rewarded PTs (PT5, PT6, and PT7 conducted on S23, S27, and S33, respectively) to test memory retrieval of OPAs for the hM4Di-CNO group. In both PT6 and PT7, the rats spent significantly more time digging the cued sand well above the chance level if the tests are done without CNO, indicating that the rats learned the OPAs and could retrieve it. However, CNO prevented memory retrieval during these PTs. ***p<0.001, one-sample t-test comparing the proportion of digging time at the cued sand well with the chance level of 16.67%. Non-rewarded PT4 (S35) which was conducted after replacing two OPAs with two NPAs (NPAs 9 and 10) in S34 for the hM4Di-CNO group. Results show that the rats spent significantly more time digging the new cued sand well above the chance level if CNO was not given before the PT, indicating that the rats learned the NPAs from S34 and could retrieve it in this PT. However, if CNO is given before the PT, the retrieval is impaired. ***p<0.001, one-sample t-test comparing the proportion of digging time at the new cued sand well with the chance level of 16.67%. (**K and L**) Latency (in seconds) before commencing digging at the correct well. Data shown as mean ± SD. (**M, N, and O**) Continuation study (S21–34) with the rescue group (n=5). The PI (mean ± SD) is shown in (**M**). PT5 and PT6 (conducted at S23 and S28, respectively) are shown in (**N**). PT7, which was conducted twice, is shown in (**O**). In S21, PI is 75.3 ± 4.5% for the six OPAs without CNO or L-lactate. For S22–28, two OPAs were replaced with two NPAs (NPAs 9 and 10). In S22, which was conducted with CNO only, PI dropped to 63.3 ± 5.6%. PT5 (**N**) confirms that the rats did not learn the NPAs 9 and 10 from S21. In S24–26, which were conducted with CNO only, PI remained similarly low (69.3 ± 4.9%, 66 ± 7.7%, and 66 ± 5.7%, respectively), indicating that the rats were not learning the NPAs 9 and 10 despite multiple sessions. In S27, which was conducted with CNO+L-lactate, PI raised to 74 ± 5.7%, suggesting that they have learned the NPAs in this session. This was confirmed by PT6 (**N**) which showed that they spent significantly more time in digging the new cued sand well above the chance level. In S29–34, the six OPAs were restored. Studies in these sessions showed that PI drops from ~77% to ~64% even for the OPAs if L-lactate is not given concurrently with CNO. Furthermore, PT7 (S34) (**O**) shows that CNO administration before PT impairs memory retrieval of existing associative schema which can be rescued by administering L-lactate concurrently. *p<0.05, **p<0.01, one-sample t-test comparing the proportion of digging time at the cued sand well with the chance level of 16.67%.

The online version of this article includes the following source data and figure supplement(s) for figure 2:

**Source data 1.** Zip file containing data for *Figure 2A–O*, *Figure 2—figure supplement 2A–D*, and *Figure 2—figure supplement 3E–G* in GraphPad Prism file format.

**Figure supplement 1.** Schema experimental design.

**Figure supplement 2.** Clozapine-*N*-oxide (CNO) application itself has no effect on paired associate (PA) learning and memory retrieval.

**Figure supplement 3.** Results of the open field test (OFT).

PA training sessions impaired PA learning. Next, we substituted CNO with IP saline and continued the training of the hM4Di-CNO group (n=8) using the six original PAs (OPAs) (*Figure 2B*). The PI gradually increased and reached 77.5 ± 3.2% after 10 training sessions (S32 in *Figure 2B*). At S34, when two NPAs (NPAs 9 and 8) were introduced, the PI was 69.6 ± 5.1%. These results suggest that, when CNO is withdrawn, the rats in the hM4Di-CNO group can learn PAs, like the control group.

## $G_i$ pathway activation in the ACC astrocytes reduces cAMP and L-lactate levels in the ACC

Astrocyte-derived L-lactate or exogenous L-lactate has been shown to confer beneficial effects in cognitive functions in several studies (*Newman et al., 2011*; *Suzuki et al., 2011*; *Wang et al., 2017*; *Harris et al., 2019*; *Vezzoli et al., 2020*; *Iqbal et al., 2022*). As hM4Di is a $G_i$-coupled receptor, its activation by CNO could lead to inhibition of adenylyl cyclase, resulting in a decreased level of cyclic adenosine monophosphate (cAMP) (*Jones et al., 2018*). cAMP in astrocytes acts as a trigger for L-lactate production (*Choi et al., 2012*; *Horvat et al., 2021a*; *Horvat et al., 2021b*; *Zhou et al., 2021*). We hypothesized that hM4Di activation in ACC astrocytes could lead to a decrease in cAMP with a consequent decrease in L-lactate level in the ACC. To confirm this, we prepared a cohort of eight rats by habituation and pretraining for PA experiments (*Figure 3A*). Then, bilateral injection of AAV8-GFAP-hM4Di-mCherry into the ACC was done in these rats. After 3 weeks, all rats were trained for two PA training sessions with six PAs. In S3, rats were given IP CNO (3 mg/kg body weight, n=4 rats) or saline (n=4 rats). After 30 min, PA training was started, and the rats were sacrificed at 60 min of CNO or saline administration. The brain was collected, and immunohistochemistry (IHC) was done to assess the cAMP level. As shown in *Figure 3B* and *Figure 3C*, cAMP was reduced in the hM4Di-expressed

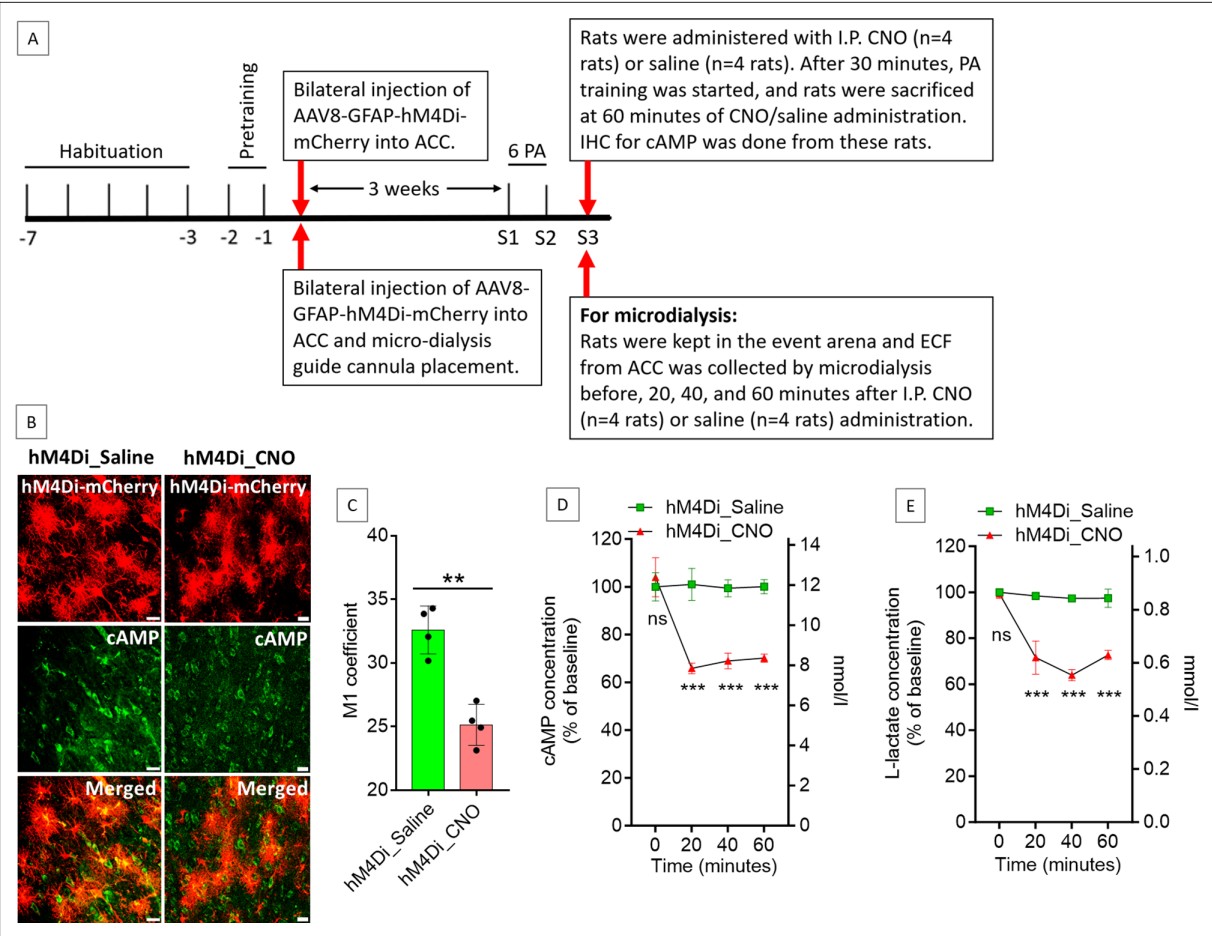

**Figure 3.** Effect of anterior cingulate cortex (ACC) astrocytic $G_i$ activation on cyclic adenosine monophosphate (cAMP) and L-lactate levels in the ACC.
(**A**) Experimental design to investigate the effect of $G_i$ activation of ACC astrocytes on cAMP and L-lactate levels. (**B and C**) CNO decreases cAMP in the hM4Di-expressed cells. (**B**) Confocal micrograph of ACC 60 min after intraperitoneal administration of saline or CNO in hM4Di-expressed rats. Scale bars: 20 μm. (**C**) Colocalization analysis showing decreased Mander's coefficient M1 (ratio of cAMP intensity colocalized with hM4Di-mCherry to total cAMP intensity) in CNO administered rats (n=4 rats in each group; 3 sections/rat). **p=0.001, unpaired Student's t-test, t=6.01, df = 6. (**D and E**) Microdialysis measurement of cAMP (**D**) and L-lactate (**E**) levels in the extracellular fluid (ECF) of ACC before, 20 min, 40 min, and 60 min after intraperitoneal saline or CNO administration in hM4Di-expressed rats (n=4 rats in each group). ns = not significant, ***p<0.001, unpaired Student's t-test.

The online version of this article includes the following source data for figure 3:

**Source data 1.** Zip file containing data for **Figure 3C–E** in GraphPad Prism file format.

cells in the CNO-treated rats compared to the saline-treated rats. Colocalization analysis (**Figure 3C**) of hM4Di-mCherry with cAMP showed decreased Mander's coefficient M1 (ratio of cAMP intensity colocalized with hM4Di-mCherry to total cAMP intensity) in the CNO-treated rats compared to the saline-treated rats (25.1±1.6 vs 32.6±1.9, respectively; t=6.01, df = 6, p=0.001, unpaired Student's t-test).

Most cells that generate cAMP, including astrocytes, can export a portion of it into the extracellular fluid (ECF) (**Stone and John, 1990**; **Rosenberg, 1992**). Extracellular cAMP levels correlate with the intracellular cAMP levels and could be used to indirectly assess the intracellular cAMP levels and therefore the activity of adenylyl cyclase in the awake, freely moving animals (**Stone and John, 1990**; **Nomikos et al., 2000**; **Klamer et al., 2005**). To detect the effect of $G_i$ activation of the ACC astrocytes on the ECF cAMP levels at different timepoints, another cohort of eight rats was prepared similarly and was given CNO (n=4) or saline (n=4) IP at S3 (**Figure 3A**). We collected ECF from the ACC by microdialysis before, 20, 40, and 60 min after the CNO or saline administration. As shown in **Figure 3D**, we observed a significant reduction in cAMP from baseline in the ACC ECF at these

timepoints after CNO injection. These results indicated that the reduction in cAMP due to astrocytic $G_i$ activation could be observed as early as 20 min after IP administration of CNO in hM4Di-expressed rats and the decrease is sustained at least until 60 min after CNO administration. We also measured the L-lactate levels in these microdialysate samples of the ACC. As shown in *Figure 3E*, we observed a significant decrease in the L-lactate level in the ACC at 20, 40, and 60 min after CNO injection compared to saline, suggesting a decreased L-lactate production from astrocytes due to $G_i$ activation.

## Administration of exogenous L-lactate can rescue the astrocytic $G_i$ pathway activation-mediated impairment in PA learning

Given that the $G_i$ activation in the ACC astrocytes decreases L-lactate levels in the ACC, we reasoned that the impaired PA learning observed in the astrocytic $G_i$ pathway-activated rats could be rescued by exogenous L-lactate administration if, indeed, the impairment was due to a decreased L-lactate level. To investigate this, we prepared another group of rats (n=7). These rats received bilateral injections of AAV8-GFAP-hM4Di-mCherry into the ACC to express hM4Di-mCherry in the astrocytes. They also received CNO 30 min before the start and 30 min after the end of each PA training session (similar to the hM4Di-CNO group). Moreover, they received exogenous L-lactate bilaterally (10 nmol, 1 µl per ACC) into the ACC 15 min after receiving CNO injections. Hereafter, this group of rats will be referred to as the 'rescue group'.

As shown in *Figure 2A*, the rescue group showed consistently higher PI than the hM4Di-CNO group. Interestingly, their PI was even higher than the control group in the middle stage (especially in S10–12) of PA learning (statistical data is given in *Supplementary file 2*). In S10–12, the PIs of the rescue group were >77%, which was achieved by the control group only at the late stage of PA training (S16–17). Overall, the findings suggested that the impaired PA learning observed in the hM4Di-CNO group was due to decreased L-lactate levels in the ACC upon astrocytic $G_i$ pathway activation, and that exogenous L-lactate not only rescues this impairment but may also reduce the number of required PA training sessions to learn the six OPAs.

## $G_i$ pathway activation in the ACC astrocytes impairs memory retrieval whereas concurrent exogenous L-lactate administration rescues the impairment

Non-rewarded probe tests (PTs) were performed at S3, S9, and S18 to test the PA memory retrieval. *Figure 2C–E* shows the results of PT1–3, respectively. In PT1 (*Figure 2C*), no rat group spent significantly more time digging the cued sand well above the chance level. In both PT2 (*Figure 2D*) and PT3 (*Figure 2E*), the control group spent significantly more time digging the cued sand well above the chance level, indicating that they learned the PAs from the previous PA training sessions and were able to retrieve it during the non-rewarded PTs (PT2: 24.1 ± 5.8%, one-sample t-test, t=3.39, df = 6, p=0.015; PT3: 38.4 ± 9.7%, one-sample t-test, t=6.34, df = 7, p<0.001).

For PT1–3 in hM4Di-CNO group (n=15), eight rats received IP saline, whereas seven rats received IP CNO 30 min before each of these PTs. As shown in *Figure 2C–E*, none of these subgroups spent more time digging the cued sand well above the chance level in any of these PTs. Later, PT5–7 (*Figure 2G–I*) were conducted for the rats of hM4Di-CNO group that underwent PA training sessions without CNO from S21 onward (n=8) (*Figure 2B*). Each of these PTs was conducted twice. One test was done with IP saline in the morning and another test was done with IP CNO in the afternoon. As shown in *Figure 2G–I*, these rats spent more time digging the cued sand well above the chance level in PT6 (23.8 ± 2.7%, one-sample t-test, t=7.61, df = 7, p<0.001) and PT7 (34.5 ± 7.7%, one-sample t-test, t=6.55, df = 7, p<0.001) when the tests were conducted without CNO, consistent with their gradual PA learning from the PA training sessions in the absence of CNO (S21–32). However, when the PTs were conducted with CNO, they did not spend more time digging the cued sand well.

In the rescue group, both PT2 and PT3 were conducted twice (*Figure 2D and E*). One test was done with only CNO in the morning and another test was done with CNO+L-lactate in the afternoon. We found that the rescue group could not retrieve PA memory if only CNO was given. However, they could retrieve PA memory if L-lactate was given concurrently with CNO as evidenced by the significantly more digging time spent in the cued sand well above the chance level (PT2: 27.8 ± 6.6%, one-sample t-test, t=4.45, df = 6, p=0.004; PT3: 32.1 ± 5%, one-sample t-test, t=8.24, df = 6, p<0.001). Taken together, these PT results suggested that $G_i$ pathway activation in the ACC astrocytes can

impair retrieval of already learned PAs, and concurrent exogenous L-lactate administration can rescue the impairment in memory retrieval.

## $G_i$ pathway activation in the ACC astrocytes impairs NPA learning despite the existence of prior associative schema whereas exogenous L-lactate administration rescues the impairment

Rats that have prior associative schema showed rapid acquisition of NPAs in a single trial (*Tse et al., 2007*; *Hasan et al., 2019*; *Liu et al., 2022*). We replaced two of the six OPAs with two NPAs (NPAs 7 and 8) at S19 (*Figure 2A*) and conducted non-rewarded PT4 (PT4) after 24 hr (*Figure 2F*). Consistent with other reports (*Tse et al., 2007*; *Hasan et al., 2019*; *Liu et al., 2022*), the control group was able to learn the NPAs from the single PA training session (PI 69.5 ± 7.1%) and retrieve it during the PT4 as evidenced by the significantly more digging time spent at the correct new cued location (percent digging time in new cued sand well 42.9 ± 12.2%, one-sample t-test, t=6.1, df = 7, p<0.001). Similarly, the hM4Di-CNO group, which did not learn the NPAs at S19 (*Figure 2A*) and showed no retrieval of these NPAs in PT4 (*Figure 2F*), was able to learn the NPAs (NPAs 9 and 8) in a single PA training session at S34 (*Figure 2B*) and retrieved the NPAs in PT8 (*Figure 2J*) when the PT was done without CNO (percent digging time in the new cued sand well 35.4±8.7, one-sample t-test, t=6.11, df = 7, p<0.001). The rescue group also learned the NPAs from the single PA training session (PI 68.1 ± 7.1%) (*Figure 2A*) and retrieve it during the PT4 (*Figure 2F*) when the test was done with CNO+L-lactate (percent digging time in the new cued sand well 36.5 ± 5%, one-sample t-test, t=10.41, df = 6, p<0.001). This result confirmed that the rescue group developed the associative schema like control group and can assimilate the NPAs into the existing schema in a single PA training session if CNO+L-lactate is given during the PA training session (S19). However, PT4 without concurrent L-lactate administration (i.e., with only CNO) showed impaired memory retrieval in the rescue group, which, together with the result of PT8 of the hM4Di-CNO group, suggested that astrocytic $G_i$ activation in the ACC impairs NPA memory retrieval and exogenous L-lactate administration can rescue the retrieval impairment.

NPA learning requires activation and retrieval of existing associative schema stored in the ACC. We reasoned that $G_i$ pathway activation in the ACC astrocytes might impair NPA learning even in rats having associative schema memory due to $G_i$ pathway activation-mediated impairment of memory retrieval from the ACC. To test this hypothesis, we used five rats from the rescue group for further study (*Figure 2M–O*). In S21 (*Figure 2M*), we checked the PI of these rats using the six OPAs without giving CNO or L-lactate. The PI was 75.3 ± 4.5%. In S22, we replaced two OPAs with two NPAs (NPAs 9 and 10) and performed PA training with CNO only. The PI dropped to 63.3 ± 5.6%. Then we performed PT5 with CNO+L-lactate (*Figure 2N*). The rats did not spend significantly more time digging the new cued sand well than the chance level. This indicated that even though these rats already had associative schema memory, they could not learn the NPAs from a single PA training session due to the ACC astrocytic $G_i$ pathway activation during the training session with NPAs.

Next, we examined whether these rats can learn the NPAs if we increase the number of training sessions. With the same PAs as in S22, we continued to do three more PA training sessions (S24–26) with CNO. As shown in *Figure 2M*, the PIs in these sessions (69.3 ± 4.9%, 66 ± 7.7%, and 66 ± 5.7%, respectively) remained similar to the PI of S22, suggesting that the rats were not learning the NPAs despite multiple training sessions. However, when we administered CNO+L-lactate at S27, the PI raised to 74 ± 5.7%. This training session was followed by PT6 (*Figure 2M*) with CNO+L-lactate. The rats spent significantly more time digging the new cued sand well above the chance level (percent digging time in the new cued sand well 29.9 ± 7.2%, one-sample t-test, t=4.12, df = 4, p=0.015), indicating that the rats learned the NPAs 9 and 10 from S27. The results suggested that exogenous administration of L-lactate can rescue the impaired NPA learning ability of the ACC astrocytic $G_i$ pathway activated rats.

Next, we investigated whether these rats can recall OPAs if ACC astrocytic $G_i$ pathway is activated, and no exogenous L-lactate is given. In S29 and S30, we checked the PI of rats by injecting both CNO and L-lactate (*Figure 2M*). Similar to the PIs in S8–17, the PIs in these two sessions were 77.3 ± 2.5% and 76.7 ± 3.7%, respectively. S31–33 were done with CNO only. In these sessions, the PIs dropped to 65.3 ± 2.7%, 63.3 ± 3%, and 64.7 ± 4%, respectively, indicating poorer performance without exogenous L-lactate, which is similar to the PIs of the hM4Di-CNO group. In S34, PT7 was conducted

twice: once with only CNO and once with CNO+L-lactate (*Figure 2O*). The rats could not retrieve the existing associative schema memory if L-lactate was not given in addition to CNO, suggesting impaired memory retrieval if astrocytic $G_i$ pathway is activated.

## CNO application itself has no effect on PA learning and memory retrieval

Although CNO had long been considered biologically inert, studies showed that it is converted to clozapine. CNO was implicated in reduced startle response to loud acoustic stimuli and clozapine-like interoceptive stimulus effects in rodents (*MacLaren et al., 2016*; *Manvich et al., 2018*). Therefore, we investigated whether CNO itself had an effect on PA learning, schema formation, and memory retrieval. Rats (n=4) were bilaterally injected with AAV8-GFAP-mCherry into the ACC. After habituation and pretraining, these rats were similarly trained for PA learning. Before 30 min and after 30 min of each PA training session, they received IP CNO. As shown in *Figure 2—figure supplement 2*, CNO did not affect PA learning, schema formation, memory retrieval, NPA learning and retrieval, or latency (time needed to commence digging at the correct well). They behaved similarly to the control group. This result is consistent with our recent study, where CNO did not affect PA learning and schema formation in rats bilaterally injected with AAV8-GFAP-mCherry into the CA1 of the hippocampus (*Liu et al., 2022*).

## Administration of CNO or CNO+L-lactate in rats expressing hM4Di in the ACC astrocytes does not induce abnormalities in an open field test

To test whether the ACC astrocytic $G_i$ activation by CNO, or the combination of $G_i$ activation and exogenous L-lactate administration, causes abnormalities in locomotion, we conducted an open field test (OFT). Rats (three groups: hM4Di-saline, hM4Di-CNO, and hM4Di-CNO+L-lactate; n=8 in each group) were prepared by injecting AAV8-GFAP-hM4Di-mCherry bilaterally into the ACC (*Figure 2—figure supplement 3*). After habituation, pretraining, and two PA training sessions, the OFT was conducted for all groups. No differences were observed in terms of the distance traveled, time spent in the central zone, or number of entries into the central zone. OFT was also performed after S8 and S17 for the hM4Di-CNO group (n=8), which showed no significant changes in these parameters.

## $G_i$ pathway activation in the ACC astrocytes reduces neuronal mitochondrial biogenesis whereas concurrent exogenous L-lactate administration rescues it

Mitochondrial dysfunction is a hallmark of numerous diseases that cause cognitive decline, for example, neurodegenerative diseases, genetic mitochondrial diseases, and aging (*Golpich et al., 2017*; *Khacho et al., 2017*). Multiple recent studies have provided striking evidence of the role of mitochondrial biogenesis in hippocampus-dependent cognitive functions (*Khacho et al., 2017*; *Liu et al., 2018*; *Han et al., 2020*; *Jacobs et al., 2021*). A recent study has demonstrated that exercise-induced L-lactate release from skeletal muscle or IP injection of L-lactate can induce hippocampal PGC-1α (peroxisome proliferator-activated receptor-gamma coactivator 1-alpha) expression and mitochondrial biogenesis in mice (*Park et al., 2021*). Recently, we observed increased expression of SIRT3, PGC-1α, and mitochondrial markers in the hippocampal neurons, along with an elevated mtDNA copy number, in anesthetized rats 1 hr after bilateral administration of exogenous L-lactate into the hippocampus (*Akter et al., 2023*). SIRT3 is known to promote mitochondrial biogenesis, reduce reactive oxygen species (ROS) production, and plays important role in learning and memory (*Fu et al., 2012*; *Ansari et al., 2017*; *Satoh et al., 2017*; *Kim et al., 2019*; *Liu et al., 2019*; *Liu et al., 2021*; *Sun et al., 2021*). PGC-1α was shown to activate SIRT3 promoter (*Sun et al., 2021*). On the other hand, SIRT3 was shown to promote PGC-1α expression (*Fu et al., 2012*), suggesting a positive feedback loop between SIRT3 and PGC-1α. As we have shown that ACC astrocytic $G_i$ activation decreases L-lactate in the ACC ECF, we hypothesized that the PGC-1α/SIRT3/mitochondrial biogenesis axis could have been downregulated in the ACC neurons in the hM4Di-CNO group of rats. *Figure 4* shows the results from the control, hM4Di-CNO, and rescue groups of rats used for schema experiments in the current study. The control rats did not receive CNO or L-lactate before being sacrificed. hM4Di-CNO group received IP CNO 1 hr before being sacrificed. Rescue group received IP CNO 75 min before and bilateral exogenous L-lactate into the ACC 60 min before being sacrificed.

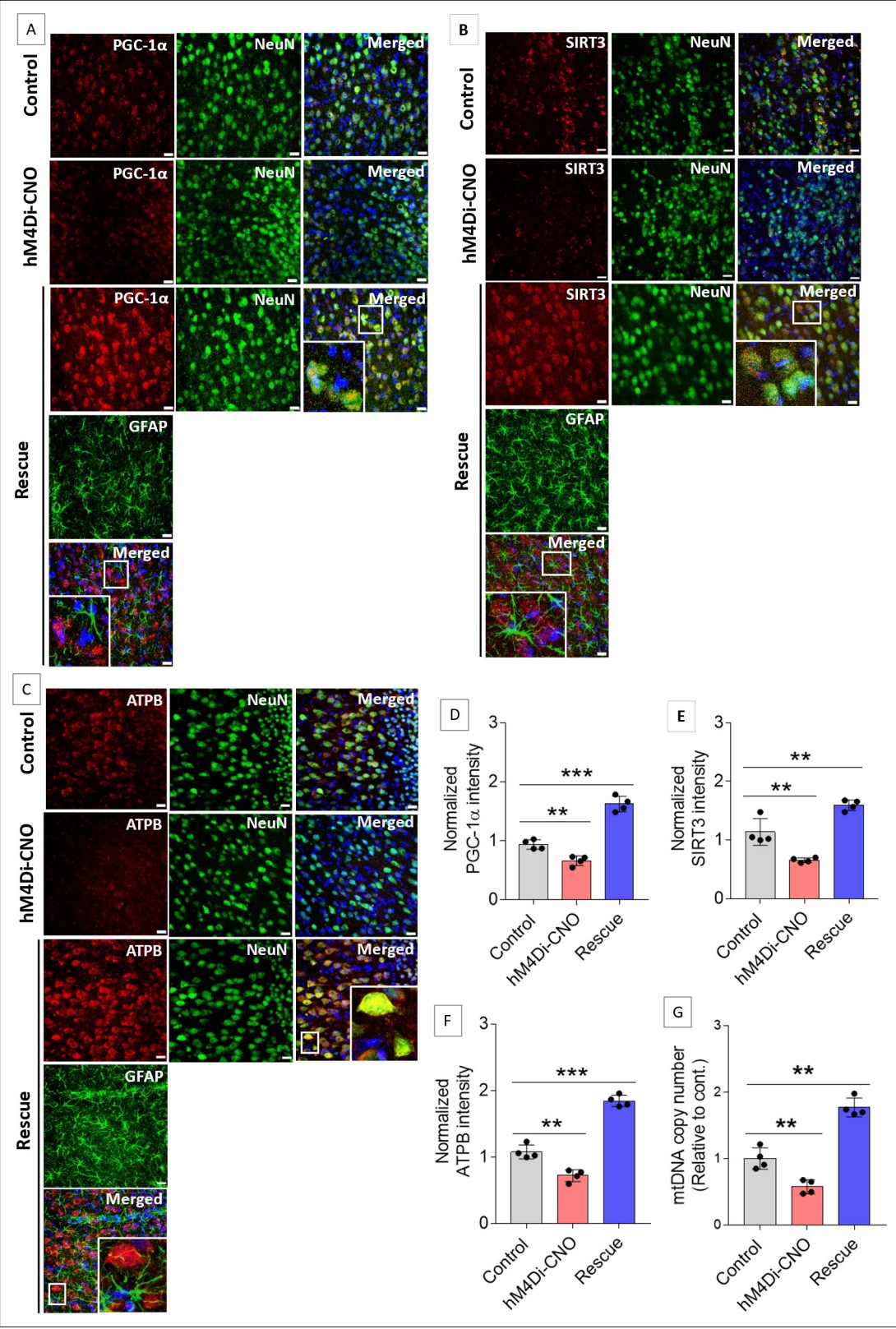

**Figure 4.** G$_i$ activation in anterior cingulate cortex (ACC) astrocytes reduces neuronal mitochondrial biogenesis whereas concurrent exogenous L-lactate administration rescues the impairment. (**A–C**) Representative confocal micrograph of PGC-1α (**A**)/SIRT3 (**B**)/ATPB (**C**) co-labeled with NeuN, glial fibrillary acidic protein (GFAP), and DAPI (4',6-diamidino-2-phenylindole) in the ACC of the control, hM4Di-CNO, and rescue groups from schema experiments. Astrocytic G$_i$ pathway activation (hM4Di-CNO group) in the ACC resulted in decreased PGC-1α/SIRT3/ATPB expression in the ACC, whereas concurrent

*Figure 4 continued on next page*

*Figure 4 continued*
exogenous L-lactate (rescue group) administration resulted in increased PGC-1α/SIRT3/ATPB expression. Scale bars: 20 µm. (**D–F**) Fluorescence intensity of PGC-1α (**D**)/SIRT3 (**E**)/ATPB (**F**) stained sections in the ACC of hM4Di-CNO and rescue groups were assessed and normalized to the control group of rats. Data shown as mean ± SD (n=4 rats per group, 3 sections/rat). **p<0.01, ***p<0.001, unpaired Student's t-test. (**G**) mtDNA copy number abundance in the ACC of control, hM4Di-CNO, and rescue groups relative to nDNA. Relative mtDNA copy number was significantly decreased in the hM4Di-CNO group, whereas it was increased in the rescue group compared to control. Data shown as mean ± SD (n=4 rats per group). **p<0.01, unpaired Student's t-test.

The online version of this article includes the following source data for figure 4:

**Source data 1.** Zip file containing data for *Figure 4D–G* in GraphPad Prism file format.

We observed a significantly decreased expression of PGC-1α, SIRT3, and ATPB (a component of mitochondrial membrane ATP synthase) in the ACC neurons of hM4Di-CNO group compared to the control group. The relative mtDNA copy number in ACC was also decreased in the hM4Di-CNO group (*Figure 4G*). On the other hand, the rescue group showed increased expression of PGC-1α, SIRT3, and ATPB in the ACC neurons as well as increased relative mtDNA copy number in the ACC, even higher than the control group, which is consistent with their faster PA learning than the control group. Together, these results revealed that the ACC astrocytic $G_i$ activation impairs neuronal mitochondrial biogenesis by decreasing ECF L-lactate levels in the ACC and that exogenous L-lactate administration rescues the impaired mitochondrial biogenesis.

## Mitochondrial biogenesis by L-lactate is dependent on MCT2 and NMDAR activity

Previous studies demonstrated that L-lactate entry into neurons is needed for its beneficial effect (*Newman et al., 2011*; *Suzuki et al., 2011*; *Wang et al., 2017*). After entry, L-lactate promotes plasticity gene expression by potentiating NMDA signaling (*Yang et al., 2014*; *Magistretti and Allaman, 2018*). We investigated whether entry into the neuron and NMDA receptor (NMDAR) activity are required for L-lactate-induced mitochondrial biogenesis (*Figure 5*). After habituation and pretraining, cannula placement was done bilaterally into the ACC of rats. After 1 week of recovery, two PA training sessions (S1 and S2) with six OPAs were conducted. To test whether L-lactate-induced neuronal mitochondrial biogenesis is dependent on MCT2, we bilaterally injected MCT2 antisense oligodeoxynucleotide (MCT2-ODN, n=8 rats, 2 nmol in 1 µl PBS per ACC) or scrambled ODN (SC-ODN, n=8 rats, 2 nmol in 1 µl PBS per ACC) into the ACC. After 11 hr, bilateral infusion of L-lactate (10 nmol, 1 µl) or ACSF (1 µl) was given into the ACC, and the rats were kept in the PA event arena. After 60 min (12 hr from MCT2-ODN or SC-ODN administration), the rats were sacrificed. As shown in *Figure 5B*, SC-ODN+L-lactate group showed a significantly increased relative mtDNA copy number compared to the SC-ODN+ACSF group (p<0.001, ANOVA followed by Tukey's multiple comparisons test). However, this effect was completely abolished in MCT2-ODN+L-lactate group, suggesting that MCT2 is required for the L-lactate-induced mitochondrial biogenesis in the ACC.

To test whether L-lactate-induced mitochondrial biogenesis is NMDAR-dependent, we used D-(-)-2-amino-5-phosphonopentanoic acid (D-APV), which is a competitive inhibitor of the glutamate binding site of NMDAR. Four groups of rats were used for this experiment: the ACSF group, which received bilateral infusion of ACSF (1 µl) into the ACC; the L-Lactate group, which received bilateral infusion of L-lactate (10 nmol, 1 µl); the D-APV group, which received D-APV (30 mM, 0.5 µl); and the D-APV+L-Lactate group, which received L-lactate infusion 15 min after D-APV. After infusion, rats were kept in the PA event arena for 60 min and then sacrificed. While the relative mtDNA copy number was significantly increased in the L-lactate group (p<0.001, ANOVA followed by Tukey's multiple comparisons test), this effect was not observed in the D-APV+L-lactate group, suggesting that NMDAR activity is required for L-lactate-induced mitochondrial biogenesis in the ACC.

## Discussion

There has been a paradigm shift in neuroscience in which animal behavior is now considered as a result arising from the coordinated activity of neurons and glia, especially astrocytes, rather than a result exclusively from neuronal activity (*Kofuji and Araque, 2021a*). A recent study has revealed that several GPCR genes are expressed in astrocytes across the CNS, whereas some GPCR genes are

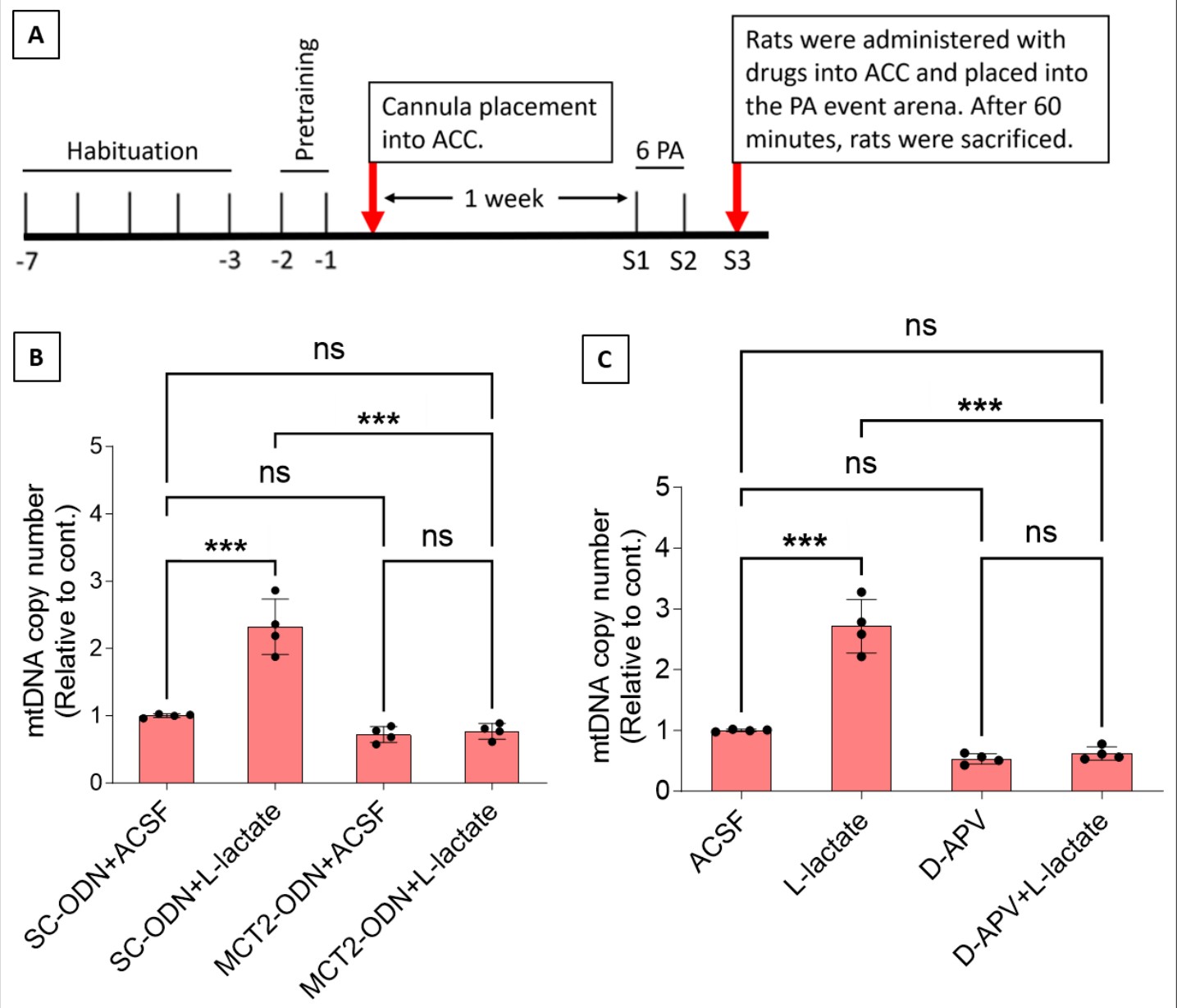

**Figure 5.** Mitochondrial biogenesis by L-lactate is dependent on monocarboxylate transporter 2 (MCT2) and NMDA receptor (NMDAR).
(**A**) Experimental design to investigate whether MCT2 and NMDAR activity are required for L-lactate-induced mitochondrial biogenesis. (**B and C**) mtDNA copy number abundance in the anterior cingulate cortex (ACC) of different rat groups relative to nDNA. Data shown as mean ± SD (n=4 rats in each group). ***p<0.001, ANOVA followed by Tukey's multiple comparisons test.

The online version of this article includes the following source data and figure supplement(s) for figure 5:

**Source data 1.** Zip file containing data for *Figure 5B–C* and *Figure 5—figure supplement 1B* in GraphPad Prism file format.

**Figure supplement 1.** Western blot of monocarboxylate transporter 2 (MCT2).

expressed in a region-specific manner (*Endo et al., 2022*). GPCRs confer astrocytes with the ability to sense synaptic activity and respond with gliotransmitters to regulate neuronal and synaptic functions (*Kofuji and Araque, 2021b*). Moreover, astrocytic responses to GPCR activation may show heterogeneity among brain regions or even within a brain region (*Kofuji and Araque, 2021b*), highlighting the complex roles of GPCRs in astrocytic functioning. L-lactate, derived primarily from astrocytes by glycogenolysis and glycolysis, has increasingly been recognized as a novel gliotransmitter that facilitates cognitive functions (*Newman et al., 2011*; *Suzuki et al., 2011*; *Wang et al., 2017*; *Magistretti*

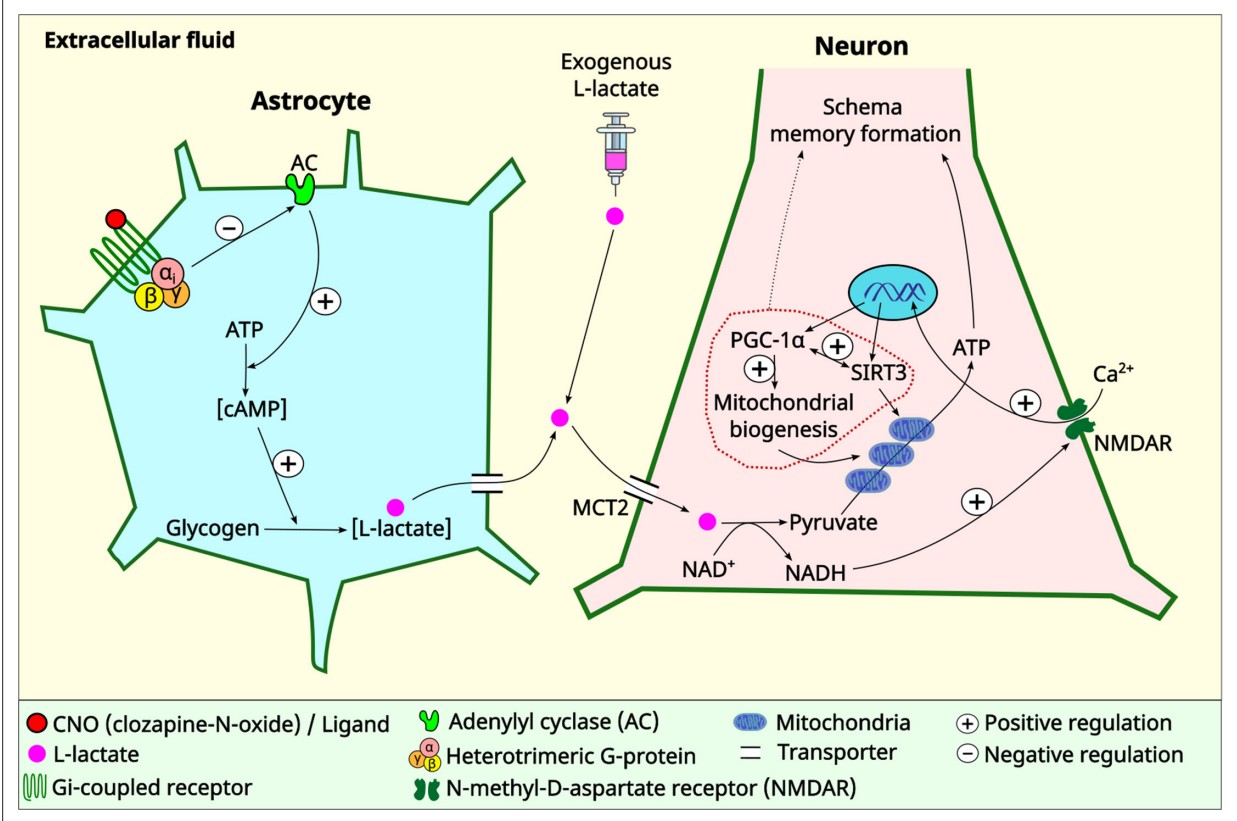

**Figure 6.** Schematic diagram showing astrocytic $G_i$ signaling and L-lactate modulating schema memory and mitochondrial biogenesis. L-lactate in the anterior cingulate cortex (ACC) is required for schema memory formation and neuronal mitochondrial biogenesis. Astrocytic $G_i$ activation results in decreased L-lactate in the ACC with consequent impairments in schema memory and neuronal mitochondrial biogenesis which could be rescued by exogenous L-lactate administration directly into the ACC. Further research is needed to establish the mechanism and the extent of the contribution of mitochondrial biogenesis in schema memory formation (dotted arrow). MCT2: monocarboxylate transporter 2.

*and Allaman, 2018*; *Harris et al., 2019*; *Vezzoli et al., 2020*). cAMP in astrocytes acts as a trigger for L-lactate production (*Choi et al., 2012*; *Horvat et al., 2021a*; *Horvat et al., 2021b*; *Zhou et al., 2021*) by promoting glycogenolysis and glycolysis (*Vardjan et al., 2018*; *Horvat et al., 2021a*; *Horvat et al., 2021b*). Our current study shows that astrocytic $G_i$ activation in the ACC decreases intracellular cAMP and ECF L-lactate levels in the ACC. Therefore, one promising explanation for the reduced L-lactate level observed in our study upon astrocytic $G_i$ activation is decreased glycogenolysis and glycolysis as a result of decreased astrocytic cAMP (*Figure 6*).

Schema is defined as a framework of knowledge. New learning occurs rapidly if it occurs against a background of established relevant schema. Rats trained with multiple flavor-place PAs develop schema that enables rapid assimilation of NPA learning (*Tse et al., 2007*; *Tse et al., 2011*; *Hasan et al., 2019*; *Liu et al., 2022*). Previous studies suggested that PA learning is hippocampus-dependent and the associative schema is stored in the ACC (*Tse et al., 2007*; *Tse et al., 2011*). In this study, we have demonstrated that ACC astrocytic $G_i$ activation impairs PA learning and schema formation, PA memory retrieval, and NPA learning and retrieval by decreasing L-lactate level in the ACC. Although we have shown that these impairments are associated with diminished expression of proteins of mitochondrial biogenesis, the precise mechanisms of how astrocytic $G_i$ activation affects neuronal functions and schema memory remain to be elucidated. We previously demonstrated that neuronal inhibition in either the hippocampus or the ACC impairs PA learning and schema formation (*Hasan et al., 2019*). In another recent study (*Liu et al., 2022*), we showed that astrocytic $G_i$ activation in the CA1 impaired PA training-associated CA1-ACC projecting neuronal activation. *Yao et al., 2023* recently showed that reduction of astrocytic lactate dehydrogenase A (an enzyme that reversibly catalyze L-lactate production from pyruvate) in the dorsomedial prefrontal cortex reduces L-lactate levels and neuronal firing frequencies, promoting depressive-like behaviors in mice. These impairments could be rescued

by L-lactate infusion. It is possible that the impairment in PA learning and schema observed in our study might have involved a similar functional consequence of reduced neuronal activity in the ACC neurons upon astrocytic $G_i$ activation.

Schema consolidation is associated with synaptic plasticity-related gene expression (such as Zif268, Arc) in the ACC (*Tse et al., 2011*). L-lactate, after entry into neurons, can be converted to pyruvate during which NADH is also produced, promoting synaptic plasticity-related gene expression by potentiating NMDA signaling in neurons (*Yang et al., 2014*; *Margineanu et al., 2018*). Furthermore, L-lactate acts as an energy substrate to fuel learning-induced de novo neuronal translation critical for long-term memory (*Descalzi et al., 2019*). On the other hand, mitochondria play crucial role in fueling local translation during synaptic plasticity (*Rangaraju et al., 2019*). Therefore, it could be hypothesized that the rescue of astrocytic $G_i$ activation-mediated impairment of schema by exogenous L-lactate could have been mediated by facilitating synaptic plasticity-related gene expression by directly fueling the protein translation, potentiating NMDA signaling, as well as increasing mitochondrial capacity for ATP production by promoting mitochondrial biogenesis. Furthermore, the potential involvement of HCAR1, a receptor for L-lactate that may regulate neuronal activity (*Bozzo et al., 2013*; *Tang et al., 2014*; *Herrera-López and Galván, 2018*; *Abrantes et al., 2019*), cannot be excluded. Future research could explore these potential mechanisms, examining the interactions among them, and determining their relative contributions to schema.

Our previous study also showed that ACC myelination is necessary for PA learning and schema formation, and that repeated PA training is associated with oligodendrogenesis in the ACC (*Hasan et al., 2019*). Oligodendrocytes facilitate fast, synchronized, and energy efficient transfer of information by wrapping axons in myelin sheath. Furthermore, they supply axons with glycolysis products, such as L-lactate, to offer metabolic support (*Fünfschilling et al., 2012*; *Lee et al., 2012*). The association of oligodendrogenesis and myelination with schema memory may suggest an adaptive response of oligodendrocytes to enhance metabolic support and neuronal energy efficiency during PA learning. Given the impairments in PA learning observed in the ACC astrocytic $G_i$-activated rats in the current study, it is reasonable to conclude that the direct metabolic support to axons provided by oligodendrocytes is not sufficient to rescue the schema impairments caused by decreased L-lactate levels upon astrocytic $G_i$ activation. On the other hand, L-lactate was shown to be important for oligodendrogenesis and myelination (*Sánchez-Abarca et al., 2001*; *Rinholm et al., 2011*; *Ichihara et al., 2017*). Therefore, it is tempting to speculate that a decrease in L-lactate level may also impede oligodendrogenesis and myelination, consequently preventing the enhanced axonal support provided by oligodendrocytes and myelin during schema learning. Recently, a study has demonstrated that upon demyelination, mitochondria move from the neuronal cell body to the demyelinated axon (*Licht-Mayer et al., 2020*). Enhancement of this axonal response of mitochondria to demyelination, by targeting mitochondrial biogenesis and mitochondrial transport from the cell body to axon, protects acutely demyelinated axons from degeneration. Given the connection between schema and increased myelination, it remains an open question whether L-lactate-induced mitochondrial biogenesis plays a beneficial role in schema through a similar mechanism. Nevertheless, our results contribute to the mounting evidence of the glial role in cognitive functions and underscores the new paradigm in which glial cells are considered as integral players in cognitive functions alongside neurons. Disruption of neurons, myelin, or astrocytes in the ACC can disrupt PA learning and schema memory. These discoveries have clinical implications, as they suggest that pathological processes involving any of these cell types can eventually result in a loss of harmony among these cells and manifest as cognitive impairments. Indeed, accumulating evidence suggests the crucial contributions of non-neuronal cells in the pathology of diverse neurodegenerative disorders, including Alzheimer's disease, Parkinson's disease, Huntington's disease, and amyotrophic lateral sclerosis (*Brandebura et al., 2023*).

In this study, we have demonstrated that ACC astrocytic $G_i$ activation impairs NPA learning even if a prior associative schema exists. This impairment is the result of impaired activation and retrieval of prior associative schema from the ACC neuronal network mediated by decreased L-lactate in ACC as exogenous L-lactate administration abolished the impairment. After an associative schema is formed in the ACC due to repeated PA training with multiple PAs, the effect of astrocytic $G_i$ activation in the ACC is different from the effect of astrocytic $G_i$ activation in the hippocampus. Whereas $G_i$ activation in the ACC leads to impairment in both PA memory retrieval and NPA learning as shown in the current study, $G_i$ activation in the hippocampus primarily affects the NPA learning but not the

memory retrieval of the previously learned PAs (*Liu et al., 2022*). This indicates that once associative schema is formed in the ACC, it becomes independent of the hippocampus, and disruption of either hippocampal neuronal (*Hasan et al., 2019*) or astrocytic (*Liu et al., 2022*) functions does not impact retrieval of the previously learned PAs. However, NPA learning in the setting of the existence of a prior associative schema can be impaired by disruption of either hippocampal functions (neuronal inhibition [*Hasan et al., 2019*] or astrocytic $G_i$ activation [*Liu et al., 2022*]) or ACC functions (neuronal inhibition [*Hasan et al., 2019*] or astrocytic $G_i$ activation [current study]), indicating that NPA learning requires simultaneous activation of both the hippocampus and ACC with optimal functioning of both neurons and astrocytes.

After entry into a neuron, L-lactate can be converted to pyruvate, during which NADH is also produced. Pyruvate can enter the mitochondria and be processed through Kreb's cycle and oxidative phosphorylation to generate ATP. Demand of ATP is high in active neurons to maintain various physiological activities, including neural plasticity and memory formation (*Magistretti and Allaman, 2018*). Although L-lactate's beneficial effect on cognitive functions has been clearly demonstrated in the current study and several other previous studies (*Newman et al., 2011*; *Suzuki et al., 2011*; *Wang et al., 2017*; *Harris et al., 2019*; *Vezzoli et al., 2020*), whether the beneficial effect is conferred by its usefulness as an energy substrate has been debated (*Díaz-García et al., 2017*). Intriguingly, a recent study has demonstrated that glucose and L-lactate metabolism are differentially engaged in neuronal fueling depending on neural computational and cognitive loads (*Dembitskaya et al., 2022*). The study showed that L-lactate is necessary for a cognitive task requiring high attentional load but is not needed for a less demanding task. This suggests that cognitive functions may exhibit varying degrees of sensitivity to L-lactate. In addition to its function as an energy source, the signaling role of L-lactate is increasingly being acknowledged. NADH produced during the conversion of L-lactate into pyruvate can induce plasticity-related gene expression by activating NMDA and/or MAPK signaling pathways (*Yang et al., 2014*; *Magistretti and Allaman, 2018*; *Margineanu et al., 2018*). The results of the current study suggest the existence of another MCT2 and NMDAR-dependent signaling role of L-lactate. We used MCT2 ODN to decrease the expression of MCT2 in the ACC and showed that MCT2 is necessary for L-lactate-induced mitochondrial biogenesis, indicating that L-lactate's entry into the neuron is required. We further investigated whether NMDAR activity is required for L-lactate-induced mitochondrial biogenesis. We used D-APV to inhibit NMDAR and found that L-lactate does not increase mtDNA copy number abundance if D-APV is given, suggesting that NMDAR activity is required for L-lactate to promote mitochondrial biogenesis. While these results suggest the involvement of MCT2 and NMDAR in the upregulation of mitochondrial biogenesis by L-lactate (*Figure 6*), we have not investigated other mechanisms and pathways modulating mitochondrial biogenesis that are either dependent or independent of MCT2 and NMDAR activity. Further studies are needed to better understand the detailed mechanisms. Moreover, it remains to be explored whether L-pyruvate, a metabolite that shares certain functional similarities with L-lactate, such as oxidative stress resistance (*Tauffenberger et al., 2019*) and neuroprotection against excitotoxicity (*Jourdain et al., 2018*), but exhibits differences in other aspects like plasticity gene expression (*Yang et al., 2014*), can effectively rescue the astrocytic $G_i$ activation-mediated schema or mitochondrial biogenesis impairments.

Our study demonstrates that ACC astrocytic $G_i$ activation resulted in a downregulation of neuronal PGC-1α and SIRT3. These proteins are key regulators of mitochondrial biogenesis and homeostasis. SIRT3, a member of sirtuin family, is a protein deacetylase that is exclusively found in mitochondria and is known to promote mitochondrial biogenesis and reduce ROS production (*Fu et al., 2012*; *Ansari et al., 2017*; *Sun et al., 2021*). Several recent studies have demonstrated that SIRT3 plays important role in learning and memory (*Kim et al., 2019*; *Liu et al., 2019*; *Liu et al., 2021*). $Sirt3^{-/-}$ mice demonstrated impaired remote memory function and decreased synaptic plasticity and neuronal number in the ACC (*Kim et al., 2019*). Another study in aged mice showed that SIRT3 overexpression can provide protection against anesthesia/surgery-induced synaptic plasticity dysfunction in the hippocampus and attenuate hippocampus-dependent cognitive decline (*Liu et al., 2021*). SIRT3 was shown to be required for the anxiolytic and cognition-enhancing effects of intermittent fasting (*Liu et al., 2019*). The study demonstrated that mice lacking SIRT3 in the hippocampal neurons have heightened anxiety, poor memory retention, and impaired long-term potentiation at hippocampal synapses. Recently, it has been demonstrated that PGC-1α overexpression in neurons can improve hippocampal neuronal function, increase ATP production, reduce oxidative stress, and attenuate

cognitive impairment after chronic cerebral hypoperfusion in mice (*Han et al., 2020*). Another study has suggested that upregulation of PGC-1α and mitochondrial biogenesis in the hippocampus enhances spatial learning and short-term memory (*Jacobs et al., 2021*). Mitochondrial dysfunction was shown to impair hippocampus-dependent learning and memory in mice (*Khacho et al., 2017*). On the other hand, another study showed that ameliorating mitochondrial dysfunction rescues carbon ion-induced hippocampal cognitive deficits (*Liu et al., 2018*). Mitochondrial dysfunction is a hallmark of numerous diseases that causes cognitive decline (*Golpich et al., 2017*). Collectively, these reports provided striking evidence of the role of mitochondrial homeostasis in cognitive functions. Therefore, the associations observed from the current study set up an interesting premise for further studies to investigate whether disruption of the L-lactate-regulated neuronal mitochondrial biogenesis plays a causal role in the cognitive impairment due to astrocytic $G_i$ activation. Based on the known functions, one might hypothesize that L-lactate-induced upregulation of PGC-1α/SIRT3/mitochondrial biogenesis could enable the neurons to generate more ATP while reducing oxidative stress during bioenergetic challenges as in cognitively demanding tasks of PA learning. In line with this, a recent study has demonstrated that L-lactate causes a mild ROS burst that induces antioxidant defenses and pro-survival pathways (*Tauffenberger et al., 2019*).

In summary, the present study illustrates that ACC astrocytic $G_i$ pathway activation impairs schema memory in rats by decreasing L-lactate levels in the ACC, which is associated with impaired mitochondrial biogenesis in neurons. These impairments can be rescued by exogenous L-lactate administration. Furthermore, we demonstrated that L-lactate-mediated neuronal mitochondrial biogenesis is dependent on MCT2 and NMDAR activity – uncovering a novel signaling mechanism of L-lactate in the brain. These results might have implications in understanding how perturbation in astrocytic functions could impair cognitive functions as well as providing the potential therapeutic targets for ameliorating such impairments.

# Materials and methods

**Key resources table**

| Reagent type (species) or resource | Designation | Source or reference | Identifiers | Additional information |
|---|---|---|---|---|
| Strain, strain background (*Rattus norvegicus*, male) | Adult Sprague- Dawley rats (250–300 g) | Laboratory Animal Services Centre, Chinese University of Hong Kong, SAR, China | | |
| Transfected construct (*Rattus norvegicus*) | AAV8-GFAP-hM4Di-mCherry | Shanghai Taitool Bioscience Co. Ltd | | |
| Transfected construct (*Rattus norvegicus*) | AAV8-GFAP-mCherry | Shanghai Taitool Bioscience Co. Ltd | | |
| Sequence-based reagent | Rat MCT2 antisense oligodeoxynucleotide (ODN), 200 nmol (HPLC purified) | Integrated DNA Technologies (IDT) | Cat #: 107968967 | |
| Sequence-based reagent | Rat Relative scrambled ODN, 200 nmol (HPLC purified) | Integrated DNA Technologies (IDT) | Cat #: 107969138 | |
| Sequence-based reagent | Rat D-loop Forward and Reverse Primer | Integrated DNA Technologies (IDT) | Cat #: 107056074 and Cat #: 107056075 | Used to measure mtDNA by real-time PCR |
| Sequence-based reagent | Rat β-actin Forward and Reverse Primer | Integrated DNA Technologies (IDT) | Cat #: 107056076 and Cat #: 107056077 | Used to measure nDNA by real-time PCR |
| Antibody | Anti-GFAP (Mouse Monoclonal) | Abcam | Cat #: ab4648 | 1:500 (IHC) |
| Antibody | Anti-NeuN (Rabbit Polyclonal) | Merck Millipore | Cat #: AB978 | 1:500 (IHC) |
| Antibody | Anti-mCherry (Chicken Polyclonal) | Abcam | Cat #: ab205402 | 1:1000 (IHC) |
| Antibody | Anti-cAMP (Rabbit Monoclonal) | Abcam | Cat #: ab134901 | 1:500 (IHC) |
| Antibody | Anti-SIRT3 (Rabbit Polyclonal) | Sigma-Aldrich | Cat #: SAB5700222 | 1: 250 (IHC) |
| Antibody | Anti-PGC-1α (Rabbit Polyclonal) | Abcam | Cat #: ab191838 | 1: 500 (IHC) |

*Continued on next page*

*Continued*

| Reagent type (species) or resource | Designation | Source or reference | Identifiers | Additional information |
|---|---|---|---|---|
| Antibody | Anti-ATPB (Mouse Monoclonal) | Abcam | Cat #: ab14730 | 1: 500 (IHC) |
| Antibody | Anti-MCT2 (Rabbit Polyclonal) | Merck Millipore | Cat #: AB3542 | 1: 500 (WB) |
| Antibody | Anti-β-actin (Mouse Monoclonal) | Immunoway | Cat #: YM3028 | 1: 5000 (WB) |
| Antibody | Alexa Flour 488 (Goat Anti-Mouse Polyclonal) | Thermo Fisher Scientific | Cat #: A11001 | 1:300 (IHC) |
| Antibody | Alexa Flour 594 (Goat Anti-Mouse Polyclonal) | Thermo Fisher Scientific | Cat #: A11032 | 1:300 (IHC) |
| Antibody | Alexa Flour 488 (Goat Anti-Rabbit Polyclonal) | Thermo Fisher Scientific | Cat #: A11034 | 1:300 (IHC) |
| Antibody | Alexa Flour 594 (Goat Anti-Mouse Polyclonal) | Thermo Fisher Scientific | Cat #: A11037 | 1:300 (IHC) |
| Antibody | Goat Anti-Rabbit Secondary Antibody, HRP, Polyclonal | Invitrogen | Cat #: 31460 | 1: 5000 (WB) |
| Antibody | Goat Anti-Mouse Secondary Antibody, HRP, Polyclonal | Invitrogen | Cat #: 62–6520 | 1: 5000 (WB) |
| Commercial assay or kit | Lactate Fluorescence Assay kit | Abcam | Cat #: ab65331 | |
| Commercial assay or kit | cAMP complete ELISA kit | Abcam | Cat #: ab133051 | |
| Commercial assay or kit | QIAamp DNA Mini Kits | QIAGEN | Cat #: 1725270 | |
| Commercial assay or kit | SsoAdvanced Universal SYBR Green Supermix | Bio-Rad | Cat #: 1725270 | |
| Commercial assay or kit | RIPA Buffer | Sigma-Aldrich | Cat #: 20-188 | |
| Commercial assay or kit | Phosphatase and protease inhibitor cocktail | Sigma-Aldrich | | |
| Commercial assay or kit | Bradford assay | Bio-Rad | Cat #: 5000205 | |
| Commercial assay or kit | Western Bright ECL HRP substrate | Advansta | Cat #: K12045-D20 | |
| Chemical compound, drug | Clozapine-*N*-oxide (CNO) dihydrochloride | Hello Bio | Cat #: HB6149 | |
| Chemical compound, drug | NaCl | Sigma-Aldrich | Cat #: S3014-1kg | |
| Chemical compound, drug | L-lactate | Sigma-Aldrich | Cat #: L-7022 | |
| Chemical compound, drug | D-(-)-2-Amino-5-Phosphonopentanoic acid (D-APV) | Sigma-Aldrich | Cat #: A8054 | |
| Chemical compound, drug | Artificial cerebrospinal fluid (ACSF) | Harvard Apparatus | Cat #: 597316 | |
| Chemical compound, drug | Dorminal 20% | Alfasan International BV | Cat #: 013003 | |
| Chemical compound, drug | Urethane | Sigma-Aldrich | Cat #: U2500-500G | |
| Software, algorithm | FIJI ImageJ | National Institutes of Health, Bethesda, MD, USA | | |
| Software, algorithm | Prism GraphPad | GraphPad Software, San Diego, CA, USA | | Version 10 |
| Software, algorithm | Excel | Microsoft | | |

*Continued on next page*

*Continued*

| Reagent type (species) or resource | Designation | Source or reference | Identifiers | Additional information |
|---|---|---|---|---|
| Other | Microdialysis guide cannula (CMA 11 elite) and probe (3 mm membrane) | CMA Inc | | Used to collect ECF from ACC for L-lactate and cAMP assay |
| Other | Stainless steel guide cannulae, OD 0.41 mm-27G/C | RWD Life Science | Cat #: 62069 | Used in drug and ODN delivery into ACC |
| Other | Dummy cannulae | RWD Life Science | Cat #: 62169 | Used in drug and ODN delivery into ACC |
| Other | Brain slicer | Braintree Scientific, Braintree, MA, USA | | Used to collect ACC from whole brain |
| Other | Stereotaxic frame | RWD | | Used to fix head of rats during surgeries |
| Other | 33-Gauge metal needle, 10 μl micro-syringe | Hamilton, NV, USA | | Used in AAV and drug delivery |
| Other | Microinjection pump | World Precision Instruments, USA | | Used in AAV delivery |

## Animal use and care

Adult male Sprague-Dawley rats weighting about 250–300 g were used in this study. All rats were housed in a standard laboratory facility (25°C, 50% humidity, 12 hr light/dark cycle with light on at 7:00 AM). All animals were supplied by the Laboratory Animal Services Centre, Chinese University of Hong Kong. The animal experimentation procedures were carried out according to the guidelines created by the Committee on Use and Care of Animals, Department of Health, Hong Kong SAR. The following are the license numbers to conduct experiments: (22-2) in DH/HT&A/8/2/5 Pt.8 and (22-3) in DH/HT&A/8/2/5 Pt.8. The approval for 'Ethical Review of Research Experiments Involving Animal Subjects' were taken by Animal Research Ethics Sub-Committee, City University of Hong Kong (References: A-0417 and A-0513). Rats were provided with food and water ad libitum except for the period of schema experiments when food restriction was applied.

## PA behavioral protocol

### PA experiment design

We used a behavioral paradigm of multiple flavor-place PAs learning as described previously (**Tse et al., 2007**). The event arena, as shown in **Figure 2—figure supplement 1A**, contains four start boxes and multiple sand wells. A flavored food pellet (flavor cue) is given in the start box, and a specific sand well (place cue) contains three more of that flavored food pellet at the bottom of it. There are multiple specific flavor-place PAs, for example beef flavor is paired with sand well number 1, strawberry flavor is paired with sand well number 2, and so on. When a rat is placed in a start box that contains a specific flavor cue, they need to use spatial memory to find out the correct sand well that contains that specific flavored food.

Our experimental setup and timeline is illustrated in **Figure 2—figure supplement 1B**. We habituated rats for 5 days (sessions –7 to –3) so that they become familiar with the event arena and learn digging sand wells. Then, we conducted pretraining for 2 days (sessions –2 and –1) to introduce them to the six original flavor-place PAs (OPAs). After that, sessions 1–18 were conducted as 4–5 sessions/ week. In each of the sessions (S) of 1, 2, 4–8, 10–17, each rat was trained with six PAs, and the normal control rats are expected to learn the flavor-place associations so that if a flavor cue is given in the start box, they should be able to find out the correct cued sand well to get more of that flavored food. S3, S9, S18 were non-rewarded PTs where rats were given a flavor cue at the start box, but the cued sand well did not contain any food pellet. After getting the flavor cue at the start box, rats were given 120 s to find out the cued sand well. PTs reflect memory retrieval. If a rat can retrieve PAs memory well, it will spend more time in digging the cued sand well. In S19, two NPAs (NPAs 7 and 8) were introduced by replacing two old PAs (PA1 and PA6, respectively). The normal control rats, using the existing schema developed from the previous sessions, are expected to learn the NPAs in this single session. This session was followed by S20 which is a non-rewarded PT, where the learning and memory

retrieval of the NPAs learned from S19 were tested. If a rat learns the NPAs introduced in S19 well, it will spend more time digging the new cued sand well.

## Performance measures in PA training sessions

*PI:* It was calculated for each rat with the following formula:

$$\text{PI} = \left[ 100 - \frac{(\textit{Total number of errors for all 6 PAs in a session for a rat} \div 6)}{5} \times 100 \right] \%$$

Errors is the number of incorrect (non-cued) sand well(s) the rat dug before digging the correct well (cued). Digging was defined as displacement of sand around sand well by rat.

## Performance measures in PT1 (S3), PT2 (S9), and PT3 (S18)

Digging time (out of 120 s) in the cued and non-cued sand wells were measured. Then proportion of time spent in digging the cued and non-cued wells in respect to the total digging time was calculated as follows:

Percentage of digging time in cued well =

$$\frac{\textit{Total time spent in digging the cued well}}{\textit{Total digging time}} \times 100\%$$

Percentage of digging time in non-cued well =

$$\frac{\textit{Total time spent in digging the noncued wells} \div 5}{\textit{Total digging time}} \times 100\%$$

## Performance measures in PT4 (S20)

Percentage of digging time in new cued well =

$$\frac{\textit{Total time spent in digging the new cued well}}{\textit{Total digging time}} \times 100\%$$

Percentage of digging time in new non-cued well =

$$\frac{\textit{Total time spent in digging the new noncued well}}{\textit{Total digging time}} \times 100\%$$

Percentage of digging time in original non-cued wells =

$$\frac{\textit{Total time spent in digging the original noncued wells} \div 4}{\textit{Total digging time}} \times 100\%$$

## Open field test

OFT was performed in a square-shaped apparatus ($80 \times 80 \times 40$ cm$^3$) to evaluate the animals' locomotor activity and anxiety-like behaviors. The rats were familiarized with the open field testing room over a period of 30–60 min for 2 consecutive days. After habituation, pretraining, and 2 PA training sessions, the OFT was conducted for three groups of rats (hM4Di-saline, hM4Di-CNO, and hM4Di-CNO+L-lactate). OFT was also performed after S8 and S17 for the hM4Di-CNO group. Thirty minutes after CNO or saline and 15 min after L-lactate injection, the rats were placed in the OFT for 5 min. Total distance (m) traveled, time (s) spent in the central zone, and the numbers of entries into the central zone for each rat were measured using Any-maze (Stoelting Co., Wood Dale, IL, USA) tracking software. The open field was cleaned with 75% ethanol between each trial. The central zone of the open field was defined by sketching a square ($40 \times 40$ cm$^2$) at the center of the apparatus.

## Stereotactic surgical procedures, viral vector injection, and CNO administration

To express hM4Di in the ACC astrocytes, AAV8-GFAP-hM4Di-mCherry was used (original viral titer $3 \times 10^{12}$ vg/ml diluted in 1:10 in PBS, Shanghai Taitool Bioscience Co. Ltd). Rats were anesthetized with

50 mg/kg sodium pentobarbital (Dorminal 20%, Alfasan International BV, Woerden, Holland) administered IP and placed in a stereotaxic frame. After exposing the skull, bilateral craniotomy was done (0.5–0.8 mm holes, 2.2–3.8 mm anterior to bregma, 0.5–1.0 mm lateral from midline). A 10 μl microsyringe (Hamilton, NV, USA) with a 33-gauge metal needle was used to perform the microinjections. We injected 400 nl of viral vector bilaterally into the ACC regions (2–3 mm ventral from the surface of the skull at the craniotomy site) with injection flow rate of 0.1 μl/min (controlled by microinjection pump, World Precision Instruments, USA) (*Figure 1—figure supplement 1*). The needle was left in place for an additional 5 min after the injection was completed. Then it was slowly withdrawn. After withdrawing the needle, the scalp was sutured, and immediate postoperative care was provided with local anesthetic (xylocaine, 2%) applied to the incision site for analgesia and allowing the rats to recover from anesthesia under a heat pad. The rats were returned to their home cage after awaking. All rats were allowed 3 weeks of rest to ensure hM4Di expression.

CNO dihydrochloride (Hello Bio, Avonmouth, UK, cat. HB6149), a synthetic ligand to activate hM4Di, was dissolved in 0.9% NaCl and was injected IP at a dose of 3 mg/kg body weight. This dose did not produce any seizure in rats.

## Chronic ACC cannulation

Rats were anesthetized with 50 mg/kg IP sodium pentobarbital administration. Stainless steel guide cannulae (double/OD 0.41 mm-27G/C, Cat #: 62069, RWD Life Science) were bilaterally positioned into the ACC region (2.2–3.8 mm anterior to bregma and 0.5–1.0 mm lateral from midline, 2 mm dorso-ventral from skull surface) (*Figure 1—figure supplement 1*). The guide cannulae were fixed to the skull with dental cement (mega PRESSNV+JET-X, megadental GmbH, Budingen, Germany). Dummy cannulae (Cat #: 62169, RWD Life Science) 0.5 mm longer than the guide cannulae were inserted into the guide cannulae to prevent blockage and reduce the risk of infection. The rats were provided with a minimum recovery period of 1 week before other experimental procedures.

Drugs were administered bilaterally into the ACC at a flow rate of 0.333 μl/min using a 33-gauge internal injecting needle. Drugs and their doses per ACC: 10 nmol L-lactate (Sigma-Aldrich, Cat #: L7022) in 1 μl ACSF (*Wang et al., 2017*); 0.5 μl of 30 mM D-APV (Sigma-Aldrich, Cat #: A8054) dissolved in ACSF (*Wang et al., 2012*). MCT2 antisense oligodeoxynucleotide (ODN) (MCT2-ODN; 5′-GACTCTGATGGCATTTCTGAG-3′) or relative scrambled ODN (SC-ODN; 5′-GGTTTACGAGTCGTCCGTAAT-3′) were dissolved in PBS pH 7.4, as described previously (*Suzuki et al., 2011*). ODNs were phosphorothioated on the three terminal bases at each end to protect against nuclease degradation. ODNs were HPLC-purified and purchased from Integrated DNA Technologies (IDT). Two nmol in 1 μl of ODNs were injected per ACC. The needle was kept in place for an additional 5 min to allow proper diffusion.

## Measurement of cAMP and L-lactate levels

To investigate the effect of ACC astrocytic $G_i$ pathway activation on cAMP and L-lactate levels in the ACC, 16 rats were habituated and pretrained for PA experiment as shown in *Figure 3A*. Then bilateral AAV8-GFAP-hM4Di-mCherry injection into the ACC was done as described before in all rats. In addition, a micro-dialysis guide cannula (CMA Inc) was inserted into the right sided ACC (2.5 mm ventral from the surface of the skull at the craniotomy site) in the rats that was used for microdialysis later (eight rats). After 3 weeks, all rats were trained for two sessions with six OPAs. For microdialysis in the next session (S3), rats were given IP CNO (3 mg/kg body weight) (n=4 rats) or saline (n=4 rats) and placed in the PA even arena. ECF from the ACC was collected before, 20, 40, and 60 min after CNO or saline administration. For collecting ECF, a microdialysis probe which is a Y-shaped catheter containing an inlet and outlet port with a fibrous, semi-permeable membrane at the bottom tip (CMA 11 elite, 3 mm membrane) was inserted into guide cannula. One fluorinated ethylene propylene (FEP) tube (ID 0.12 mm, CMA Inc) was connected to the inlet port and another FEP tube was connected to outlet port. Through inlet FEP tube, artificial cerebrospinal fluid (ACSF, Harvard Apparatus, Cat #: 597316) was infused into the ACC to maintain artificial neurotransmitter concentration gradient. Through the outlet tube, ECF from the ACC was collected by micro-infusion pump (WPI). For IHC staining of cAMP in S3, rats were given IP CNO (3 mg/kg body weight) (n=4 rats) or saline (n=4 rats). After 30 min, PA training was started, and the rats were sacrificed at 60 min of CNO or saline administration.

The dialysate collected from the ACC were kept at –80°C until further use. cAMP complete ELISA kit (Abcam, USA, Cat #: ab133051) was used to determine the cAMP concentration in ACC dialysate according to the manufacturer's protocol. Lactate Fluorescence Assay kit (Abcam, USA, Cat #: ab65331) was used to determine the L-lactate concentration from the same ACC dialysate according to the manufacturer's protocol.

## IHC and confocal microscopy

After completing experiments, rats were anesthetized by urethane (1.5 g/kg, IP) and perfused transcardially with ice-cold PBS for approximately 5 min and then perfused with 4% paraformaldehyde (PFA). The whole brain was taken out and postfixed in 4% PFA overnight at 4°C and cryoprotected in 30% sucrose dissolved in 1× PBS for an additional 3 days at 4°C. The brains were then stored in OCT medium at –80°C until further use. For IHC, each brain was sectioned at 40 μm using cryostat (Leica, USA) and processed as free-floating sections. Six to eight sections were selected for staining per rat. Sections were incubated with blocking solution of Triton X-100 (0.3% [vol/vol]) and 10% normal goat serum in 0.01 M PBS for 1 hr at room temperature after a brief wash. Then sections were incubated with primary antibodies in blocking solution for overnight at 4°C. In the following day, slices were washed three times (5 min each) and incubated with targeted Alexa Flour secondary antibodies (1: 300) in DAPI (4',6-diamidino-2-phenylindole) for 2 hr at room temperature. Then the sections were mounted into microscopic slides (Epredia SuperFrost Plus Adhesion Microscopic Slides) and covered with coverslips (Eprdia Cover Slip) along with fluorescent mounting medium (DAKO). The imaging was done by inverted laser scanning confocal microscope (LSM 880; Carl Zeiss, Oberkochen, Germany). The confocal images for quantitative analysis were acquired under 20× or 40× oil-immersion objectives. The ratio between the intensity of fluorescence and area of analysis ($mm^2$) was calculated using FIJI ImageJ software and taken as quantitative expression of targeted immunofluorescence.

## Relative mitochondrial DNA content quantification

After completing experiments, rats were anesthetized by urethane (1.5 g/kg, IP) followed by decapitation. Brain was sectioned on an anodized aluminum brain slicer (Braintree Scientific, Braintree, MA, USA) and the ACC were dissected from the sections and kept at –80°C. Total genomic DNA was extracted from ACC using QIAamp DNA Mini Kits (Cat #: 51304) according to the manufacturer's protocol. Genomic DNA quality and quantity were checked with NanoDrop 1000 Spectrophotometer (Thermo Scientific). The DNA sample was stored at –20°C until further use. Quantitative real-time PCR was performed with the SsoAdvanced Universal SYBR Green Supermix (Cat #: 1725270) using Applied Biosystems QuantStudio 3 Real-Time PCR Systems. β-Actin gene and mitochondrial D-loop were used as nuclear DNA (nDNA) and mtDNA, respectively, to investigate the abundance of mtDNA relative to nDNA, as described previously (*Akter et al., 2023*). The primer sequences (*Branda et al., 2002*; *Chou et al., 2007*) and reaction mixture protocol are given in *Supplementary file 3*. Thermal cycling was done according to the SsoAdvanced Universal SYBR Green Supermix protocol. DNA from each rat was amplified as triplicate. After obtaining both mtDNA and nDNA Ct values from Real Time PCR software, Ct values were averaged from triplicates of each rat. To determine the mtDNA content relative to the nDNA, the following equation (*Rooney et al., 2015*) was used:

$$\text{Relative mitochondrial DNA content} = 2 \times 2^{(\text{nDNA Ct} - \text{mtDNA Ct})}$$

## Western blot analysis

The samples were homogenized in mixture (100:1) of RIPA buffer (150 mM NaCl, 50 mM Tris pH 7.4, 1% Triton X-100, 0.1% SDS, 1% sodium deoxycholate) and phosphatase and protease inhibitor cocktail (Sigma-Aldrich). Homogenates were then centrifuged at 14,000 × $g$ for 30 min at 4°C. The supernatants were collected carefully as total protein and then Bradford assay (Bio-Rad, Cat #: 5000205) was used to determine the protein concentration. Using sodium dodecyl sulphate-polyacrylamide gel electrophoresis, 20 μg of proteins were separated, and then transferred to a polyvinylidene fluoride membrane. Then the membrane was incubated with 5% non-fat milk in TBST (containing 0.1% Tween 20) for 1 hr followed by incubation with primary antibodies with reference concentration for overnight at 4°C. Next day, membranes were washed for three times (5 min each) with TBST and incubated

for 2 hr with horseradish peroxidase coupled secondary goat anti-rabbit or anti-mouse IgG (1:5000, Invitrogen) in TBST. Then membranes were washed three times (5 min each). Western Bright ECL HRP substrate (Advansta, Inc, Cat #: K12045-D20) was used to visualize the blot. Images were captured and processed by Chemidoc Touch Imaging System (Bio-Rad) and quantified by FIJI ImageJ software (National Institutes of Health, Bethesda, MD, USA). Expression of target proteins were normalized with that of β-actin level.

## Data analysis

Data analyses were done with Prism v.10 (GraphPad Software, La Jolla, CA, USA) or MS Excel. Data are presented as mean ± SD as appropriate. Comparisons of continuous data were done with two-tailed Student's t-test where appropriate. Image analysis was done with ImageJ. Figures were generated with Prism v.10 and Inkscape.

## Acknowledgements

This work was funded by the General Research Fund (GRF) of the Research Grants Council of Hong Kong (11103721, 11102820, and 11100018), the National Natural Science Foundation of China (NSFC) and RGC Joint Research Scheme (3171101014, N_CityU114/17), Health@InnoHK funding support from the Innovation Technology Commission of the Hong Kong SAR (CityU 9445909). This work was also supported by City University of Hong Kong Neuroscience Research Infrastructure Grant (9610211) and Centre for Biosystems, Neuroscience, and Nanotechnology Grant (9360148).

## Additional information

### Funding

| Funder | Grant reference number | Author |
|---|---|---|
| General Research Fund (GRF) of the Research Grants Council of Hong Kong | 11103721 | Ying Li |
| General Research Fund (GRF) of the Research Grants Council of Hong Kong | 11102820 | Ying Li |
| General Research Fund (GRF) of the Research Grants Council of Hong Kong | 11100018 | Ying Li |
| National Natural Science Foundation of China (NSFC) and RGC Joint Research Scheme | 3171101014 | Ying Li |
| National Natural Science Foundation of China (NSFC) and RGC Joint Research Scheme | N_CityU114/17 | Ying Li |
| Health@InnoHK funding support from the Innovation and Technology Commission of the Hong Kong SAR | CityU 9445909 | Ying Li |
| City University of Hong Kong Neuroscience Research Infrastructure Grant | 9610211 | Ying Li |

| Funder | Grant reference number | Author |
|---|---|---|
| Center for Biosystems, Neuroscience, and Nanotechnology Grant | 9360148 | Ying Li |

The funders had no role in study design, data collection and interpretation, or the decision to submit the work for publication.

## Author contributions

Mastura Akter, Conceptualization, Data curation, Software, Formal analysis, Investigation, Visualization, Methodology, Writing – original draft; Mahadi Hasan, Aruna Surendran Ramkrishnan, Zafar Iqbal, Xianlin Zheng, Zhongqi Fu, Zhuogui Lei, Investigation; Anwarul Karim, Formal analysis, Visualization, Methodology; Ying Li, Conceptualization, Resources, Supervision, Funding acquisition, Methodology, Project administration, Writing - review and editing

## Author ORCIDs

Mastura Akter http://orcid.org/0000-0003-0007-6075
Zafar Iqbal http://orcid.org/0000-0003-2604-7184
Anwarul Karim http://orcid.org/0000-0002-5795-6674
Ying Li http://orcid.org/0000-0003-3683-9695

## Ethics

The animal experimentation procedures were carried out according to the guidelines created by the Committee on Use and Care of Animals, Department of Health, Hong Kong SAR. The following are the license numbers to conduct experiments: (22-2) in DH/HT&A/8/2/5 Pt.8 and (22-3) in DH/HT&A/8/2/5 Pt.8. The approval for 'Ethical Review of Research Experiments Involving Animal Subjects' were taken by Animal Research Ethics Sub-Committee, City University of Hong Kong (References: A-0417 and A-0513).

## Decision letter and Author response

Decision letter https://doi.org/10.7554/eLife.85751.sa1
Author response https://doi.org/10.7554/eLife.85751.sa2

# Additional files

## Supplementary files

• Supplementary file 1. Comparison of performance index of control vs. hM4D$_i$-CNO group.

• Supplementary file 2. Comparison of performance index of control vs. rescue group.

• Supplementary file 3. Primer sequences and preparation of 20 µl reaction mixture for real-time PCR.

• MDAR checklist

## Data availability

All data generated or analysed during this study are included in the manuscript, source data files, and supplementary files.

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
