## [Editor Report]

This is an important study that investigates the role of astrocytic Gi signaling in the anterior cingulate cortex in the modulation of extracellular L-lactate level and consequently impairment in flavor-place paired associates learning. The evidence supporting the authors' main conclusions is convincing. Additionally, the study provides compelling evidence to suggest the molecular mechanism by which astrocyte-produced L-lactate may influence mitochondrial biogenesis in neurons, thereby affecting schema memory. This study expands our understanding of how disruptions in astrocytic functions can impair cognitive processing and, therefore, could have clinical relevance.

---

## [Decision Letter]

**Decision letter after peer review:**

Thank you for submitting your article "Astrocytic L-lactate signaling in the anterior cingulate cortex is essential for schema memory and neuronal mitochondrial biogenesis" for consideration by *eLife*. Your article has been reviewed by 3 peer reviewers, and the evaluation has been overseen by a Reviewing Editor and Ma-Li Wong as the Senior Editor. The reviewers have opted to remain anonymous.

Essential revisions:

We collectively agree that the present study provides valuable insight into the role of L-lactate in schema memory formation and neuronal mitochondrial biogenesis in rat anterior cingulate cortex. However, following a consultation session between the Reviewing Editor and the Reviewers various methodological concerns were raised. Based on these, we have enumerated the following recommendations that we encourage the authors to carefully address. In addition, the assessment from each of the reviewers is included below. With your revision, please include a point-by-point response to each reviewer's comment.

1) As currently employed, the use of Gi DREADD doesn't allow for disambiguating effects on learning vs retrieval. See comments from Reviewer #2. The authors should modify their Gi approach to distinguish learning vs retrieval effects.

2) Both Reviewers #1 and #2 feel that additional evidence is required to support diminished biogenesis in mitochondria of ACC neurons. Reviewer #1 has suggested using more specific methods for interfering with MCT2.

3) Reviewer #3 raised an important point regarding the role of L-lactate signaling in schema memory formation, and in particular the role of astrocytic-derived L-lactate in this process. We think that the authors should address this important limitation either experimentally or via discussion in the text.

*Reviewer #1 (Recommendations for the authors):*

What is, if any, according to the authors the physiological activator of Gi signaling in astrocytes?

The authors mention the importance of oligodendrocytes and myelination in schema memory formation in the ACC. Work by Nave's group has clearly shown that oligodendrocytes also produce the L-lactate required for axonal integrity. Can the authors comment on this point?

Did the authors consider using L-pyruvate to counteract the effect of Gi activation?

*Reviewer #2 (Recommendations for the authors):*

– Please indicate the specificity for ACC targeting (virus injection maps, cannula placement for drug injections).

– What is the reason for injecting CNO 30 min post-session? Line 319 and 434. How long each behavioral session lasts? CNO levels peak within 30 min after i.p. injection and last about 90-120 min.

– For more precise data interpretation please provide additional information on how CNO and CNO/L-lactate injections are affecting locomotor activity in Gi-DREADDs expressing rats.

– It appears that CNO and L-lactate have been injected before retrieval sessions PT1, PT2, and PT3. Because of that it is hard to differentiate the acute effect of CNO on learning versus the effect on memory retrieval. To claim the memory effect an additional group should be added in which CNO and CNO/L-lactate injections will be stopped after session 8.

– CNO effect on cAMP and lactate level at S3 of PA (Figure 3) is demonstrated by collecting probes for microdialysis -20, 40, and 60 min after the single CNO application. In addition, all samples for analysis illustrated in Figures 4 and 5 were collected 60 min after CNO administration. Should an additional time point be selected to measure effects in the absence of CNO (2-3 hrs later)?

– Would it be possible to provide a range of cAMP and L-lactate concentrations (in mM) obtained with microdialysis? Often hidden in papers, when represented as normalized value, but very useful information to have for results interpretation (within the physiological range, comparable to other studies, unique property of selected brain region, etc).

– Please provide at least some data on how neurons in ACC are affected by astrocytic Gi-DREADDs activation (changes in evoked or spontaneous synaptic activity, AMPA/NMDA ratio, ca^2+^ signal…) and if those effects are linked to mitochondrial biogenesis (blocked by 4-CIN?). Would administration of 4-CIN impair learning or prevent L-lactate-mediating rescue of PA learning?

*Reviewer #3 (Recommendations for the authors):*

– Some aspects of data analysis need to be clarified. For instance, in imaging experiments, it is not clear what the sample size is. It is understood that number of animals used in the individual experiment represents the number of tissue slices analysed? One animal is one tissue slice.

– The title of the paper should be relativized. There is no direct evidence that L-lactate "signalling" is involved in schema memory formation. Furthermore, there is no direct evidence that L-lactate originates from astrocytes (e.g. MCT1/4 blockers should be used to provide this evidence). The work provides only indirect evidence on the involvement of astroglial L-lactate.

---

## [Author Response]

Essential revisions:We collectively agree that the present study provides valuable insight into the role of L-lactate in schema memory formation and neuronal mitochondrial biogenesis in rat anterior cingulate cortex. However, following a consultation session between the Reviewing Editor and the Reviewers various methodological concerns were raised. Based on these, we have enumerated the following recommendations that we encourage the authors to carefully address. In addition, the assessment from each of the reviewers is included below. With your revision, please include a point-by-point response to each reviewer's comment.1) As currently employed, the use of Gi DREADD doesn't allow for disambiguating effects on learning vs retrieval. See comments from Reviewer #2. The authors should modify their Gi approach to distinguish learning vs retrieval effects.

We have now added 8 more rats to the hM4Di-CNO group (i.e., the group with astrocytic Gi activation) to clarify the memory retrieval. These rats underwent flavor-place paired associate (PA) training similar to the previously described rats (n=7) of this group, that is they received CNO 30 minutes before and 30 minutes after the PA training sessions (S1-2, S4-8, S10-17). However, contrasting to the previous rats of this group which received CNO before PTs (PT1, PT2, PT3), we omitted the CNO (instead administered I.P. saline) selectively on these PTs conducted at the early, middle, and late stage of PA training, as suggested by the reviewer. These newly added rats did not show memory retrieval in these PTs, suggesting that the rats were not learning the PAs from the PA training sessions. See Figure 2C-E, where this subgroup is denoted as *hM4Di-CNO (Saline)*.

We then continued more PA training sessions (S21 onwards, Figure 2B) for these rats without CNO. They gradually learned the PAs. PTs (PT5, PT6, PT7; Figure 2G-I) were done during this continuation phase of PA training; once without CNO (i.e., with I.P. saline instead), and another one with CNO. As seen in Figure 2H and 1I, they retrieved the memory when PT6 and PT7 were done without CNO. However, if these PTs were done with CNO, they could not retrieve the memory. Together these results suggest that ACC astrocytic Gi activation by CNO during PT can impair memory retrieval in rats which have already learned the PAs.

As shown in Figure 2B, we replaced two original PAs with two new PAs (NPA 9 and 10) at S34. This was followed by PT8 (S35). As seen in Figure 2J, these rats retrieved the NPA memory if the PT is done without CNO. However, they could not retrieve the NPA memory if the PT was done with CNO. This result suggests that ACC astrocytic Gi activation by CNO during PT can impair NPA memory retrieval.

In summary, these data show that astrocytic Gi activation in the ACC can impair PA memory retrieval. We have integrated this new data and results in the revised manuscript.

2) Both Reviewers #1 and #2 feel that additional evidence is required to support diminished biogenesis in mitochondria of ACC neurons. Reviewer #1 has suggested using more specific methods for interfering with MCT2.

In the revised manuscript, we have conducted the experiment using MCT2 antisense oligodeoxynucleotide (ODN) as suggested.

To test whether the L-lactate-induced neuronal mitochondrial biogenesis is dependent on MCT2, we bilaterally injected MCT2 antisense oligodeoxynucleotide (MCT2-ODN, n=8 rats, 2 nmol in 1 μl PBS per ACC) or scrambled ODN (SC-ODN, n=8 rats, 2 nmol in 1 μl PBS per ACC) into the ACC. After 11 hours, bilateral infusion of L-lactate (10 nmol, 1 μl) or ACSF (1 μl) was given into the ACC and the rats were kept in the PA event arena. After 60 minutes (12 hours from MCT2-ODN or SC-ODN administration), the rats were sacrificed. As shown in Figure 5B, SC-ODN+L-lactate group showed significantly increased relative mtDNA copy number compared to the SC-ODN+ACSF group (*p*<0.001, ANOVA followed by Tukey's multiple comparisons test). However, this effect was completely abolished in MCT2-ODN+L-lactate group, suggesting that MCT2 is required for the L-lactate-induced mitochondrial biogenesis in the ACC.

We have integrated this new data and results in the revised manuscript.

3) Reviewer #3 raised an important point regarding the role of L-lactate signaling in schema memory formation, and in particular the role of astrocytic-derived L-lactate in this process. We think that the authors should address this important limitation either experimentally or via discussion in the text.

We have modified the title of the manuscript as suggested by the reviewer #3 as follows:

“Astrocyte and L-lactate in the anterior cingulate cortex modulate schema memory and neuronal mitochondrial biogenesis”.

In addition, in the revised manuscript, we have mentioned the limitations and provided further discussions on the potential mechanisms that could underlie the L-lactate’s role in schema memory as well as potential mechanism that could result in a reduction in L-lactate in the ACC upon astrocytic Gi activation. The relevant discussions in the revised manuscript are given below:

“L-lactate, derived primarily from astrocytes by glycogenolysis and glycolysis, has increasingly been recognized as a novel gliotransmitter that facilitates cognitive functions (Newman et al., 2011; Suzuki et al., 2011; Wang et al., 2017; Magistretti and Allaman, 2018; Harris et al., 2019; Vezzoli et al., 2020). cAMP in astrocytes acts as a trigger for L-lactate production (Choi et al., 2012; Horvat, Muhič, et al., 2021; Horvat, Zorec, et al., 2021; Zhou et al., 2021) by promoting glycogenolysis and glycolysis (Vardjan et al., 2018; Horvat, Muhič, et al., 2021; Horvat, Zorec, et al., 2021). Our current study shows that astrocytic G_i_ activation in the ACC decreases intracellular cAMP and ECF L-lactate levels in the ACC. Therefore, one promising explanation for the reduced L-lactate level observed in our study upon astrocytic G_i_ activation is decreased glycogenolysis and glycolysis as a result of decreased astrocytic cAMP.”

“In this study, we have demonstrated that ACC astrocytic G_i_ activation impairs PA learning and schema formation, PA memory retrieval, and NPA learning and retrieval by decreasing L-lactate level in the ACC. Although we have shown that these impairments are associated with diminished expression of proteins of mitochondrial biogenesis, the precise mechanisms of how astrocytic G_i_ activation affects neuronal functions and schema memory remain to be elucidated. We previously demonstrated that neuronal inhibition in either the hippocampus or the ACC impairs PA learning and schema formation (Hasan et al., 2019). In another recent study (Liu et al., 2022), we showed that astrocytic G_i_ activation in the CA1 impaired PA training-associated CA1-ACC projecting neuronal activation. Yao *et al.* recently showed that reduction of astrocytic lactate dehydrogenase A (an enzyme that reversibly catalyze L-lactate production from pyruvate) in the dorsomedial prefrontal cortex reduces L-lactate levels and neuronal firing frequencies, promoting depressive-like behaviors in mice (Yao et al., 2023). These impairments could be rescued by L-lactate infusion. It is possible that the impairment in PA learning and schema observed in our study might have involved a similar functional consequence of reduced neuronal activity in the ACC neurons upon astrocytic G_i_ activation.”

“Schema consolidation is associated with synaptic plasticity-related gene expression (such as Zif268, Arc) in the ACC (Tse et al., 2011). L-lactate, after entry into neurons, can be converted to pyruvate during which NADH is also produced, promoting synaptic plasticity-related gene expression by potentiating NMDA signaling in neurons (Yang et al., 2014; Margineanu et al., 2018). Furthermore, L-lactate acts as an energy substrate to fuel learning-induced de novo neuronal translation critical for long-term memory (Descalzi et al., 2019). On the other hand, mitochondria play crucial role in fueling local translation during synaptic plasticity (Rangaraju et al., 2019). Therefore, it could be hypothesized that the rescue of astrocytic G_i_ activation-mediated impairment of schema by exogenous L-lactate could have been mediated by facilitating synaptic plasticity-related gene expression by directly fueling the protein translation, potentiating NMDA signaling, as well as increasing mitochondrial capacity for ATP production by promoting mitochondrial biogenesis. Furthermore, the potential involvement of HCAR1, a receptor for L-lactate that may regulate neuronal activity (Bozzo et al., 2013; Tang et al., 2014; Herrera-López and Galván, 2018; Abrantes et al., 2019), cannot be excluded. Future research could explore these potential mechanisms, examining the interactions among them, and determining their relative contributions to schema.”

“Our previous study also showed that ACC myelination is necessary for PA learning and schema formation, and that repeated PA training is associated with oligodendrogenesis in the ACC (Hasan et al., 2019). Oligodendrocytes facilitate fast, synchronized, and energy efficient transfer of information by wrapping axons in myelin sheath. Furthermore, they supply axons with glycolysis products, such as L-lactate, to offer metabolic support (Fünfschilling et al., 2012; Lee et al., 2012). The association of oligodendrogenesis and myelination with schema memory may suggest an adaptive response of oligodendrocytes to enhance metabolic support and neuronal energy efficiency during PA learning. Given the impairments in PA learning observed in the ACC astrocytic G_i_-activated rats in the current study, it is reasonable to conclude that the direct metabolic support to axons provided by oligodendrocytes is not sufficient to rescue the schema impairments caused by decreased L-lactate levels upon astrocytic G_i_ activation. On the other hand, L-lactate was shown to be important for oligodendrogenesis and myelination (Sánchez-Abarca et al., 2001; Rinholm et al., 2011; Ichihara et al., 2017). Therefore, it is tempting to speculate that a decrease in L-lactate level may also impede oligodendrogenesis and myelination, consequently preventing the enhanced axonal support provided by oligodendrocytes and myelin during schema learning. Recently, a study has demonstrated that upon demyelination, mitochondria move from the neuronal cell body to the demyelinated axon (Licht-Mayer et al., 2020). Enhancement of this axonal response of mitochondria to demyelination, by targeting mitochondrial biogenesis and mitochondrial transport from the cell body to axon, protects acutely demyelinated axons from degeneration. Given the connection between schema and increased myelination, it remains an open question whether L-lactate-induced mitochondrial biogenesis plays a beneficial role in schema through a similar mechanism. Nevertheless, our results contribute to the mounting evidence of the glial role in cognitive functions and underscores the new paradigm in which glial cells are considered as integral players in cognitive functions alongside neurons.”

“In addition to its function as an energy source, the signaling role of L-lactate is increasingly being acknowledged. NADH produced during the conversion of L-lactate into pyruvate can induce plasticity-related gene expression by activating NMDA and/or MAPK signaling pathways (Yang et al., 2014; Magistretti and Allaman, 2018; Margineanu et al., 2018). The results of the current study suggest the existence of another MCT2 and NMDAR-dependent signaling role of L-lactate. We used MCT2 ODN to decrease the expression of MCT2 in the ACC and showed that MCT2 is necessary for L-lactate-induced mitochondrial biogenesis, indicating that L-lactate’s entry into the neuron is required. We further investigated whether NMDAR activity is required for L-lactate-induced mitochondrial biogenesis. We used D-APV to inhibit NMDAR and found that L-lactate does not increase mtDNA copy number abundance if D-APV is given, suggesting that NMDAR activity is required for L-lactate to promote mitochondrial biogenesis. While these results suggest the involvement of MCT2 and NMDAR in the upregulation of mitochondrial biogenesis by L-lactate, we have not investigated other mechanisms and pathways modulating mitochondrial biogenesis that are either dependent or independent of MCT2 and NMDAR activity. Further studies are needed to better understand the detailed mechanisms. Moreover, it remains to be explored whether L-pyruvate, a metabolite that shares certain functional similarities with L-lactate, such as oxidative stress resistance (Tauffenberger et al., 2019) and neuroprotection against excitotoxicity (Jourdain et al., 2018), but exhibits differences in other aspects like plasticity gene expression (Yang et al., 2014), can effectively rescue the astrocytic G_i_ activation-mediated schema or mitochondrial biogenesis impairments.”

Reviewer #1 (Recommendations for the authors):What is, if any, according to the authors the physiological activator of Gi signaling in astrocytes?

Although DREADDs have been used to manipulate astrocytic GPCR signaling for quite some time now, a comprehensive knowledge of the astrocytic GPCRs, their ligands, and downstream mechanism under physiological or pathological conditions is still limited. While few Gi-coupled astrocytic GPCRs and their physiological activator (e.g., Drd2 by dopamine (Peterson et al., 2015; Zhu et al., 2018)) have been identified, astrocytic responses to GPCR activation may involve different downstream mechanism than the mechanism used by the same receptor in other cell types. For instance, the endocannabinoid receptor CB1R expressed in neurons are mainly G_αi/o_ coupled receptors. Conversely, astrocytic CB1R have been suggested to be mainly coupled to G_αq/11_ proteins (Covelo et al., 2021). Moreover, GPCRs show heterogeneity among brain regions or even within a brain region (Kofuji and Araque, 2021). A recent study (Endo et al., 2022) has identified that several GPCR genes (*Gpr37l1*, *S1pr1*, *Ntsr2*, *Ednrb*, *Smo*, *Adora2b*, *Olfr287*, *Gpr146*, *Agtrap*, *Fzd1*, *Fzd9*, and *Npr2*) are expressed in astrocytes across the CNS, whereas some GPCRs are expressed in a region specific manner (e.g., *Prokr2* in olfactory bulb, *Drd2* in striatum), highlighting the crucial and complex roles of GPCR in astrocytic functioning. The physiological activators, the downstream effectors, and well as the functional consequences of their activation/inactivation are yet to be thoroughly investigated in a region-specific manner. Given the emerging role of astrocytes in regulating diverse brain functions and the growing interest of the researchers in astrocyte biology, we anticipate that these knowledge gaps will be addressed by the research community in future. We have briefly mentioned about astrocytic GPCR in the Discussion section of the manuscript as follows:

“A recent study has revealed that several GPCR genes are expressed in astrocytes across the CNS, whereas some GPCR genes are expressed in a region-specific manner (Endo et al., 2022). GPCRs confer astrocytes with the ability to sense synaptic activity and respond with gliotransmitters to regulate neuronal and synaptic functions (Kofuji and Araque, 2021). Moreover, astrocytic responses to GPCR activation may show heterogeneity among brain regions or even within a brain region (Kofuji and Araque, 2021), highlighting the complex roles of GPCRs in astrocytic functioning.”

The authors mention the importance of oligodendrocytes and myelination in schema memory formation in the ACC. Work by Nave's group has clearly shown that oligodendrocytes also produce the L-lactate required for axonal integrity. Can the authors comment on this point?

We have discussed this and relevant issues in the revised manuscript as follows:

“Our previous study also showed that ACC myelination is necessary for PA learning and schema formation, and that repeated PA training is associated with oligodendrogenesis in the ACC (Hasan et al., 2019). Oligodendrocytes facilitate fast, synchronized, and energy efficient transfer of information by wrapping axons in myelin sheath. Furthermore, they supply axons with glycolysis products, such as L-lactate, to offer metabolic support (Fünfschilling et al., 2012; Lee et al., 2012). The association of oligodendrogenesis and myelination with schema memory may suggest an adaptive response of oligodendrocytes to enhance metabolic support and neuronal energy efficiency during PA learning. Given the impairments in PA learning observed in the ACC astrocytic G_i_-activated rats in the current study, it is reasonable to conclude that the direct metabolic support to axons provided by oligodendrocytes is not sufficient to rescue the schema impairments caused by decreased L-lactate levels upon astrocytic G_i_ activation. On the other hand, L-lactate was shown to be important for oligodendrogenesis and myelination (Sánchez-Abarca et al., 2001; Rinholm et al., 2011; Ichihara et al., 2017). Therefore, it is tempting to speculate that a decrease in L-lactate level may also impede oligodendrogenesis and myelination, consequently preventing the enhanced axonal support provided by oligodendrocytes and myelin during schema learning. Recently, a study has demonstrated that upon demyelination, mitochondria move from the neuronal cell body to the demyelinated axon (Licht-Mayer et al., 2020). Enhancement of this axonal response of mitochondria to demyelination, by targeting mitochondrial biogenesis and mitochondrial transport from the cell body to axon, protects acutely demyelinated axons from degeneration. Given the connection between schema and increased myelination, it remains an open question whether L-lactate-induced mitochondrial biogenesis plays a beneficial role in schema through a similar mechanism. Nevertheless, our results contribute to the mounting evidence of the glial role in cognitive functions and underscores the new paradigm in which glial cells are considered as integral players in cognitive functions alongside neurons. Disruption of neurons, myelin, or astrocytes in the ACC can disrupt PA learning and schema memory.”

Did the authors consider using L-pyruvate to counteract the effect of Gi activation?

Both L-lactate and L-pyruvate support the neuronal energy requirements and are involved in different signaling pathways. L-lactate application to primary neuronal cultures of the mouse neocortex or injection of L-lactate in the sensory motor cortex of anesthetized adult mice stimulated plasticity gene (Arc, c-Fos and Zif268) expression and this increased expression was NMDAR mediated. However, L-pyruvate application was unable to replace L-lactate’s effect in this context (Yang et al., 2014). Moreover, L-lactate is also able to protect neurons against excitotoxicity through a complex metabolic pathway involving the synthesis of pyruvate, the release of ATP (via pannexin opening), and the subsequent activation of the ATP-dependent potassium channel KATP. Interestingly, this L-Lactate-induced neuroprotection pathway is mimicked by L-pyruvate (Jourdain et al., 2018). In addition, both L-lactate and L-pyruvate provide oxidative stress resistance (Tauffenberger et al., 2019). These studies suggested that L-lactate and L-pyruvate share similarities and dissimilarities in their functions. Therefore, it is of interest whether L-pyruvate can rescue the schema and mitochondrial biogenesis impairments due to astrocytic G_i_ activation. Unfortunately, we have not performed this experiment and leaving it for future studies. We have now mentioned this important point in the Discussion section of the revised manuscript as follows:

“Moreover, it remains to be explored whether L-pyruvate, a metabolite that shares certain functional similarities with L-lactate, such as oxidative stress resistance (Tauffenberger et al., 2019) and neuroprotection against excitotoxicity (Jourdain et al., 2018), but exhibits differences in other aspects like plasticity gene expression (Yang et al., 2014), can effectively rescue the astrocytic G_i_ activation-mediated schema or mitochondrial biogenesis impairments.”

Reviewer #2 (Recommendations for the authors):– Please indicate the specificity for ACC targeting (virus injection maps, cannula placement for drug injections).

We have now indicated this in the manuscript with Figure 1—figure supplement 1.

– What is the reason for injecting CNO 30 min post-session? Line 319 and 434. How long each behavioral session lasts? CNO levels peak within 30 min after i.p. injection and last about 90-120 min.

Each PA training session (six PAs) lasts for approximately 60 mins for a rat. The CNO injected before session is intended to activate the astrocytic Gi pathway in ACC during the PA training session. The CNO injected 30 mins after the training session in intended to extend the duration of Gi activation for few hours beyond the training session.

PA learning is hippocampus-dependent. However, it becomes hippocampus-independent as memory consolidation occurs in the ACC (Tse et al., 2007; Tse et al., 2011). The critical time period during which the ACC encoding and consolidation of the PA memory from hippocampus in respect to the PA training sessions occurs is not clear. However, if hippocampal lesion is done 3 hours after (compared to 48 hours after) the single session training for NPA learning, rats cannot retrieve the NPA memory later during PT (Tse et al., 2007), suggesting that the assimilation of NPA into the ACC is still dependent on hippocampal integrity at 3 hours after the training session. In the current study, we did not directly interfere with hippocampal functions, rather expressed the hM4Di in the ACC. Therefore, administration of CNO after the training sessions was intended to activate the ACC astrocytic Gi pathway for extended time period after the training session during which the memory encoding/consolidation into the ACC might still be occurring from hippocampal memory traces. The results from our study suggest that astrocytic Gi activation during and early hours after the PA training results in impaired schema. It might be of interest in future studies to further dissect the critical timepoints for successful schema consolidation into the ACC such as by activating the ACC astrocytic Gi pathway only during the training session, or only after the training session at different timepoints.

– For more precise data interpretation please provide additional information on how CNO and CNO/L-lactate injections are affecting locomotor activity in Gi-DREADDs expressing rats.

To test whether ACC astrocytic G_i_ activation by CNO, or the combination of G_i_ activation and exogenous L-lactate administration causes abnormalities in locomotion, we conducted open field test (OFT). Rats (three groups: hM4Di-Saline, hM4Di-CNO, and hM4Di-CNO+L-lactate; n=8 in each group) were prepared by injecting AAV8-GFAP-hM4Di-mCherry bilaterally into the ACC (Figure 2—figure supplement 3). After habituation, pretraining, and 2 PA training sessions, the OFT was conducted for all groups. No differences were observed in terms of the distance traveled, time spent in the central zone, or number of entries into the central zone. OFT was also performed after S8 and S17 for the hM4Di-CNO group (n=8), which showed no significant changes in these parameters.

– It appears that CNO and L-lactate have been injected before retrieval sessions PT1, PT2, and PT3. Because of that it is hard to differentiate the acute effect of CNO on learning versus the effect on memory retrieval. To claim the memory effect an additional group should be added in which CNO and CNO/L-lactate injections will be stopped after session 8.

We have now provided data clarifying that astrocytic Gi activation in the ACC can impair PA memory retrieval. Please see the authors’ response to the public review by the Reviewer #2.

– CNO effect on cAMP and lactate level at S3 of PA (Figure 3) is demonstrated by collecting probes for microdialysis -20, 40, and 60 min after the single CNO application. In addition, all samples for analysis illustrated in Figures 4 and 5 were collected 60 min after CNO administration. Should an additional time point be selected to measure effects in the absence of CNO (2-3 hrs later)?

We agree that it would be informative to investigate at later timepoints also to see how long the effect of single dose of CNO lasts in terms of cAMP, L-lactate, and mitochondrial biogenesis proteins. In the PA training, we administered CNO twice (before and after each training session) for activating astrocytic Gi for a longer period than single dose CNO. Therefore, regarding the cAMP and L-lactate level, we were primarily interested in investigating the earlier time points when we can detect changes as the PA training starts at 30 mins after CNO injection. As we detected a decrease in cAMP and L-lactate at 20 mins of CNO injection, this confirms that L-lactate level is already reduced when the PA training starts. To add a later timepoint for Figure 4 is not feasible as these rats were sacrificed at the mentioned timepoints. Similarly, we would need to add many animals for additional timepoint for Figure 5. Considering all these, we are not proceeding with adding additional timepoint, although we realize the importance of such information.

– Would it be possible to provide a range of cAMP and L-lactate concentrations (in mM) obtained with microdialysis? Often hidden in papers, when represented as normalized value, but very useful information to have for results interpretation (within the physiological range, comparable to other studies, unique property of selected brain region, etc).

Indeed, such data are very useful. We have now added a second y-axis in the figures (see Figure 3D,E) to represent the concentrations detected by the cAMP and L-lactate assays.

– Please provide at least some data on how neurons in ACC are affected by astrocytic Gi-DREADDs activation (changes in evoked or spontaneous synaptic activity, AMPA/NMDA ratio, ca^2+^ signal…) and if those effects are linked to mitochondrial biogenesis (blocked by 4-CIN?). Would administration of 4-CIN impair learning or prevent L-lactate-mediating rescue of PA learning?

We thank the reviewer for raising these important points.

In a recent study (Liu et al., 2022), we showed that astrocytic Gi activation in the CA1 of hippocampus impaired PA training-associated CA1-ACC projecting neuronal activation (as indicated by decreased c-Fos and ca^2+^). A recently published study (Yao et al., 2023) showed that lactate dehydrogenase A (LDHA) in astrocytes, which reversibly catalyze the L-lactate production from pyruvate, is critical for the regulation of L-lactate production. L-lactate levels and neuronal firing frequencies in the dmPFC were reduced if astrocytic LDHA was reduced, promoting depressive-like behaviors. These impairments could be rescued by L-lactate infusion. Knockdown of MCT2 in the dmPFC reduces neuronal firing frequencies in the dmPFC and promote depressive-like behavior mimicking the effects of reduced L-lactate production by astrocytic LDHA deficiency. Moreover, these impairments in the presence of MCT2 knockdown cannot be rescued by exogenous L-lactate. It is possible that ACC astrocytic Gi activation in our study might have reduced neuronal activation in the ACC due to decreased L-lactate level which might not be rescued by exogenous L-lactate if L-lactate transport is blocked by 4-CIN or MCT2 knockdown. In line with this, entry of L-lactate into the neurons most likely to be necessary for the beneficial effect of L-lactate on schema similar to several other studies with different paradigms (Newman et al., 2011; Suzuki et al., 2011; Netzahualcoyotzi and Pellerin, 2020; Yu et al., 2021; Yao et al., 2023).

Reviewer #3 (Recommendations for the authors):– Some aspects of data analysis need to be clarified. For instance, in imaging experiments, it is not clear what the sample size is. It is understood that number of animals used in the individual experiment represents the number of tissue slices analysed? One animal is one tissue slice.

We thank the reviewer for pointing this out. We have provided further clarification for data analysis. The n numbers provided in the images represents rat number. Three sections per rat were used for quantifying intensity of immunostaining.

– The title of the paper should be relativized. There is no direct evidence that L-lactate "signalling" is involved in schema memory formation. Furthermore, there is no direct evidence that L-lactate originates from astrocytes (e.g. MCT1/4 blockers should be used to provide this evidence). The work provides only indirect evidence on the involvement of astroglial L-lactate.

We thank the reviewer for this suggestion and revised the title accordingly as follows:

“Astrocyte and L-lactate in the anterior cingulate cortex modulate schema memory and neuronal mitochondrial biogenesis”.

References

Abrantes, H. d. C., Briquet, M., Schmuziger, C., Restivo, L., Puyal, J., Rosenberg, N., Rocher, A.-B., Offermanns, S., and Chatton, J.-Y. (2019). The Lactate Receptor HCAR1 Modulates Neuronal Network Activity through the Activation of Gα and Gβγ Subunits. *The Journal of Neuroscience*, *39*(23), 4422-4433. https://doi.org/10.1523/jneurosci.2092-18.2019

Akter, M., Ma, H., Hasan, M., Karim, A., Zhu, X., Zhang, L., and Li, Y. (2023). Exogenous L-lactate administration in rat hippocampus increases expression of key regulators of mitochondrial biogenesis and antioxidant defense [Original Research]. *Frontiers in Molecular Neuroscience*, *16*. https://doi.org/10.3389/fnmol.2023.1117146

Bozzo, L., Puyal, J., and Chatton, J.-Y. (2013). Lactate Modulates the Activity of Primary Cortical Neurons through a Receptor-Mediated Pathway. *PLoS One*, *8*(8), e71721. https://doi.org/10.1371/journal.pone.0071721

Choi, H. B., Gordon, G. R., Zhou, N., Tai, C., Rungta, R. L., Martinez, J., Milner, T. A., Ryu, J. K., McLarnon, J. G., Tresguerres, M., Levin, L. R., Buck, J., and MacVicar, B. A. (2012). Metabolic communication between astrocytes and neurons via bicarbonate-responsive soluble adenylyl cyclase. *Neuron*, *75*(6), 1094-1104. https://doi.org/10.1016/j.neuron.2012.08.032

Covelo, A., Eraso-Pichot, A., Fernández-Moncada, I., Serrat, R., and Marsicano, G. (2021). CB1R-dependent regulation of astrocyte physiology and astrocyte-neuron interactions. *Neuropharmacology*, *195*, 108678. https://doi.org/https://doi.org/10.1016/j.neuropharm.2021.108678

Descalzi, G., Gao, V., Steinman, M. Q., Suzuki, A., and Alberini, C. M. (2019). Lactate from astrocytes fuels learning-induced mRNA translation in excitatory and inhibitory neurons. *Communications Biology*, *2*(1), 247. https://doi.org/10.1038/s42003-019-0495-2

Endo, F., Kasai, A., Soto, J. S., Yu, X., Qu, Z., Hashimoto, H., Gradinaru, V., Kawaguchi, R., and Khakh, B. S. (2022). Molecular basis of astrocyte diversity and morphology across the CNS in health and disease. *Science*, *378*(6619), eadc9020. https://doi.org/10.1126/science.adc9020

Fünfschilling, U., Supplie, L. M., Mahad, D., Boretius, S., Saab, A. S., Edgar, J., Brinkmann, B. G., Kassmann, C. M., Tzvetanova, I. D., Möbius, W., Diaz, F., Meijer, D., Suter, U., Hamprecht, B., Sereda, M. W., Moraes, C. T., Frahm, J., Goebbels, S., and Nave, K.-A. (2012). Glycolytic oligodendrocytes maintain myelin and long-term axonal integrity. *Nature*, *485*(7399), 517-521. https://doi.org/10.1038/nature11007

Harris, R. A., Lone, A., Lim, H., Martinez, F., Frame, A. K., Scholl, T. J., and Cumming, R. C. (2019). Aerobic Glycolysis Is Required for Spatial Memory Acquisition But Not Memory Retrieval in Mice. *eNeuro*, *6*(1). https://doi.org/10.1523/ENEURO.0389-18.2019

Hasan, M., Kanna, M. S., Jun, W., Ramkrishnan, A. S., Iqbal, Z., Lee, Y., and Li, Y. (2019). Schema-like learning and memory consolidation acting through myelination. *FASEB J*, *33*(11), 11758-11775. https://doi.org/10.1096/fj.201900910R

Herrera-López, G., and Galván, E. J. (2018). Modulation of hippocampal excitability via the hydroxycarboxylic acid receptor 1. *Hippocampus*, *28*(8), 557-567. https://doi.org/https://doi.org/10.1002/hipo.22958

Horvat, A., Muhič, M., Smolič, T., Begić, E., Zorec, R., Kreft, M., and Vardjan, N. (2021). Ca^2+^ as the prime trigger of aerobic glycolysis in astrocytes. *Cell Calcium*, *95*, 102368. https://doi.org/https://doi.org/10.1016/j.ceca.2021.102368

Horvat, A., Zorec, R., and Vardjan, N. (2021). Lactate as an Astroglial Signal Augmenting Aerobic Glycolysis and Lipid Metabolism [Review]. *Frontiers in Physiology*, *12*. https://doi.org/10.3389/fphys.2021.735532

Ichihara, Y., Doi, T., Ryu, Y., Nagao, M., Sawada, Y., and Ogata, T. (2017). Oligodendrocyte Progenitor Cells Directly Utilize Lactate for Promoting Cell Cycling and Differentiation. *J Cell Physiol*, *232*(5), 986-995. https://doi.org/10.1002/jcp.25690

Iqbal, Z., Liu, S., Lei, Z., Ramkrishnan, A. S., Akter, M., and Li, Y. (2023). Astrocyte L-Lactate Signaling in the ACC Regulates Visceral Pain Aversive Memory in Rats. *Cells*, *12*(1), 26. https://www.mdpi.com/2073-4409/12/1/26

Jourdain, P., Rothenfusser, K., Ben-Adiba, C., Allaman, I., Marquet, P., and Magistretti, P. J. (2018). Dual action of L-Lactate on the activity of NR2B-containing NMDA receptors: from potentiation to neuroprotection. *Sci Rep*, *8*(1), 13472. https://doi.org/10.1038/s41598-018-31534-y

Kofuji, P., and Araque, A. (2021). G-Protein-Coupled Receptors in Astrocyte-Neuron Communication. *Neuroscience*, *456*, 71-84. https://doi.org/10.1016/j.neuroscience.2020.03.025

Lee, Y., Morrison, B. M., Li, Y., Lengacher, S., Farah, M. H., Hoffman, P. N., Liu, Y., Tsingalia, A., Jin, L., Zhang, P. W., Pellerin, L., Magistretti, P. J., and Rothstein, J. D. (2012). Oligodendroglia metabolically support axons and contribute to neurodegeneration. *Nature*, *487*(7408), 443-448. https://doi.org/10.1038/nature11314

Licht-Mayer, S., Campbell, G. R., Canizares, M., Mehta, A. R., Gane, A. B., McGill, K., Ghosh, A., Fullerton, A., Menezes, N., Dean, J., Dunham, J., Al-Azki, S., Pryce, G., Zandee, S., Zhao, C., Kipp, M., Smith, K. J., Baker, D., Altmann, D., Anderton, S. M., Kap, Y. S., Laman, J. D., Hart, B. A. t., Rodriguez, M., Watzlawick, R., Schwab, J. M., Carter, R., Morton, N., Zagnoni, M., Franklin, R. J. M., Mitchell, R., Fleetwood-Walker, S., Lyons, D. A., Chandran, S., Lassmann, H., Trapp, B. D., and Mahad, D. J. (2020). Enhanced axonal response of mitochondria to demyelination offers neuroprotection: implications for multiple sclerosis. *Acta Neuropathologica*, *140*(2), 143-167. https://doi.org/10.1007/s00401-020-02179-x

Liu, S., Wong, H. Y., Xie, L., Iqbal, Z., Lei, Z., Fu, Z., Lam, Y. Y., Ramkrishnan, A. S., and Li, Y. (2022). Astrocytes in CA1 modulate schema establishment in the hippocampal-cortical neuron network. *BMC Biol*, *20*(1), 250. https://doi.org/10.1186/s12915-022-01445-6

Magistretti, P. J., and Allaman, I. (2018). Lactate in the brain: from metabolic end-product to signalling molecule. *Nat Rev Neurosci*, *19*(4), 235-249. https://doi.org/10.1038/nrn.2018.19

Margineanu, M. B., Mahmood, H., Fiumelli, H., and Magistretti, P. J. (2018). L-Lactate Regulates the Expression of Synaptic Plasticity and Neuroprotection Genes in Cortical Neurons: A Transcriptome Analysis. *Front Mol Neurosci*, *11*, 375. https://doi.org/10.3389/fnmol.2018.00375

Netzahualcoyotzi, C., and Pellerin, L. (2020). Neuronal and astroglial monocarboxylate transporters play key but distinct roles in hippocampus-dependent learning and memory formation. *Progress in Neurobiology*, *194*, 101888. https://doi.org/https://doi.org/10.1016/j.pneurobio.2020.101888

Newman, L. A., Korol, D. L., and Gold, P. E. (2011). Lactate produced by glycogenolysis in astrocytes regulates memory processing. *PLoS One*, *6*(12), e28427. https://doi.org/10.1371/journal.pone.0028427

Park, J., Kim, J., and Mikami, T. (2021). Exercise-Induced Lactate Release Mediates Mitochondrial Biogenesis in the Hippocampus of Mice via Monocarboxylate Transporters. *Front Physiol*, *12*, 736905. https://doi.org/10.3389/fphys.2021.736905

Peterson, S. M., Pack, T. F., and Caron, M. G. (2015). Receptor, Ligand and Transducer Contributions to Dopamine D2 Receptor Functional Selectivity. *PLoS One*, *10*(10), e0141637. https://doi.org/10.1371/journal.pone.0141637

Rangaraju, V., Lauterbach, M., and Schuman, E. M. (2019). Spatially Stable Mitochondrial Compartments Fuel Local Translation during Plasticity. *Cell*, *176*(1), 73-84.e15. https://doi.org/10.1016/j.cell.2018.12.013

Rinholm, J. E., Hamilton, N. B., Kessaris, N., Richardson, W. D., Bergersen, L. H., and Attwell, D. (2011). Regulation of oligodendrocyte development and myelination by glucose and lactate. *J Neurosci*, *31*(2), 538-548. https://doi.org/10.1523/JNEUROSCI.3516-10.2011

Sánchez-Abarca, L. I., Tabernero, A., and Medina, J. M. (2001). Oligodendrocytes use lactate as a source of energy and as a precursor of lipids. *Glia*, *36*(3), 321-329. https://doi.org/10.1002/glia.1119

Suzuki, A., Stern, S. A., Bozdagi, O., Huntley, G. W., Walker, R. H., Magistretti, P. J., and Alberini, C. M. (2011). Astrocyte-neuron lactate transport is required for long-term memory formation. *Cell*, *144*(5), 810-823.

Tang, F., Lane, S., Korsak, A., Paton, J. F. R., Gourine, A. V., Kasparov, S., and Teschemacher, A. G. (2014). Lactate-mediated glia-neuronal signalling in the mammalian brain. *Nature Communications*, *5*(1), 3284. https://doi.org/10.1038/ncomms4284

Tauffenberger, A., Fiumelli, H., Almustafa, S., and Magistretti, P. J. (2019). Lactate and pyruvate promote oxidative stress resistance through hormetic ROS signaling. *Cell Death Dis*, *10*(9), 653. https://doi.org/10.1038/s41419-019-1877-6

Tse, D., Langston, R. F., Kakeyama, M., Bethus, I., Spooner, P. A., Wood, E. R., Witter, M. P., and Morris, R. G. (2007). Schemas and memory consolidation. *Science*, *316*(5821), 76-82. https://doi.org/10.1126/science.1135935

Tse, D., Takeuchi, T., Kakeyama, M., Kajii, Y., Okuno, H., Tohyama, C., Bito, H., and Morris, R. G. (2011). Schema-dependent gene activation and memory encoding in neocortex. *Science*, *333*(6044), 891-895. https://doi.org/10.1126/science.1205274

Vardjan, N., Chowdhury, H. H., Horvat, A., Velebit, J., Malnar, M., Muhič, M., Kreft, M., Krivec, Š. G., Bobnar, S. T., Miš, K., Pirkmajer, S., Offermanns, S., Henriksen, G., Storm-Mathisen, J., Bergersen, L. H., and Zorec, R. (2018). Enhancement of Astroglial Aerobic Glycolysis by Extracellular Lactate-Mediated Increase in cAMP [Original Research]. *Frontiers in Molecular Neuroscience*, *11*. https://doi.org/10.3389/fnmol.2018.00148

Vezzoli, E., Cali, C., De Roo, M., Ponzoni, L., Sogne, E., Gagnon, N., Francolini, M., Braida, D., Sala, M., Muller, D., Falqui, A., and Magistretti, P. J. (2020). Ultrastructural Evidence for a Role of Astrocytes and Glycogen-Derived Lactate in Learning-Dependent Synaptic Stabilization. *Cereb Cortex*, *30*(4), 2114-2127. https://doi.org/10.1093/cercor/bhz226

Wang, J., Tu, J., Cao, B., Mu, L., Yang, X., Cong, M., Ramkrishnan, A. S., Chan, R. H. M., Wang, L., and Li, Y. (2017). Astrocytic l-Lactate Signaling Facilitates Amygdala-Anterior Cingulate Cortex Synchrony and Decision Making in Rats. *Cell Rep*, *21*(9), 2407-2418. https://doi.org/10.1016/j.celrep.2017.11.012

Yang, J., Ruchti, E., Petit, J. M., Jourdain, P., Grenningloh, G., Allaman, I., and Magistretti, P. J. (2014). Lactate promotes plasticity gene expression by potentiating NMDA signaling in neurons. *Proc Natl Acad Sci U S A*, *111*(33), 12228-12233. https://doi.org/10.1073/pnas.1322912111

Yao, S., Xu, M.-D., Wang, Y., Zhao, S.-T., Wang, J., Chen, G.-F., Chen, W.-B., Liu, J., Huang, G.-B., Sun, W.-J., Zhang, Y.-Y., Hou, H.-L., Li, L., and Sun, X.-D. (2023). Astrocytic lactate dehydrogenase A regulates neuronal excitability and depressive-like behaviors through lactate homeostasis in mice. *Nature Communications*, *14*(1), 729. https://doi.org/10.1038/s41467-023-36209-5

Yu, X., Zhang, R., Wei, C., Gao, Y., Yu, Y., Wang, L., Jiang, J., Zhang, X., Li, J., and Chen, X. (2021). MCT2 overexpression promotes recovery of cognitive function by increasing mitochondrial biogenesis in a rat model of stroke. *Anim Cells Syst (Seoul)*, *25*(2), 93-101. https://doi.org/10.1080/19768354.2021.1915379

Zhou, Z., Okamoto, K., Onodera, J., Hiragi, T., Andoh, M., Ikawa, M., Tanaka, K. F., Ikegaya, Y., and Koyama, R. (2021). Astrocytic cAMP modulates memory via synaptic plasticity. *Proc Natl Acad Sci U S A*, *118*(3), e2016584118. https://doi.org/10.1073/pnas.2016584118

Zhu, J., Hu, Z., Han, X., Wang, D., Jiang, Q., Ding, J., Xiao, M., Wang, C., Lu, M., and Hu, G. (2018). Dopamine D2 receptor restricts astrocytic NLRP3 inflammasome activation via enhancing the interaction of β-arrestin2 and NLRP3. *Cell Death Differ*, *25*(11), 2037-2049. https://doi.org/10.1038/s41418-018-0127-2